# Taxonomic, phylogenetic, and functional diversity of mollusk death assemblages in coral reef and seagrass sediments from two shallow gulfs in Western Cuban Archipelago

Rosely Peraza-Escarrá[1,2]*, Maickel Armenteros[2], Raúl Fernández-Garcés[3], Adolfo Gracia[2]

1 Posgrado en Ciencias del Mar y Limnología, Universidad Nacional Autónoma de México, Ciudad de México, México, 2 Instituto de Ciencias del Mar y Limnología, Universidad Nacional Autónoma de México, Ciudad de México, México, 3 Centro de Estudios Ambientales de Cienfuegos, Ciudad de Cienfuegos, Cuba

* roselyperaza2014@gmail.com

**Data Availability Statement:** All relevant data are within the manuscript and its Supporting Information files.

## Abstract

Mollusk death assemblages are formed by shell remnants deposited in the surficial mixed layer of the seabed. Diversity patterns in tropical marine habitats still are understudied; therefore, we aimed to investigate the taxonomic, phylogenetic, and functional diversity of mollusk death assemblages at regional and local scales in coral reef sands and seagrass meadows. We collected sediment samples at 11 sites within two shallow gulfs in the Northwestern Caribbean Sea and Southeastern Gulf of Mexico. All the shells were counted and identified to species level and classified into biological traits. We identified 7113 individuals belonging to 393 species (290 gastropods, 94 bivalves, and nine scaphopods). Diversity and assemblage structure showed many similarities between gulfs given their geological and biogeographical commonalities. Reef sands had higher richness than seagrasses likely because of a more favorable balance productivity-disturbance. Reef sands were dominated by epifaunal herbivores likely feeding on microphytobenthos and bysally attached bivalves adapted to intense hydrodynamic regime. In seagrass meadows, suspension feeders dominated in exposed sites and chemosynthetic infaunal bivalves dominated where oxygen replenishment was limited. Time averaging of death assemblages was likely in the order of 100 years, with stronger effects in reef sands compared to seagrass meadows. Our research provides evidence of the high taxonomic, phylogenetic, and functional diversity of mollusk death assemblages in tropical coastal sediments as result of the influence of scale-related processes and habitat type. Our study highlights the convenience of including phylogenetic and functional traits, as well as dead shells, for a more complete assessment of mollusk biodiversity.

**Funding:** RPE received support from a PhD scholarship (CVU_1080567) granted by the Consejo Nacional de Humanidades, Ciencias y Tecnologías (CONAHCYT). MA received support from a postdoctoral fellowship (CVU_982475) also granted by CONAHCYT. Additional funds were obtained by The Ocean Foundation through the "Proyecto Tres Golfos". The funders had no role in study design, data collection and analysis, decision to publish, or preparation of the manuscript.

**Competing interests:** The authors have declared that no competing interests exist.

## Introduction

Death assemblages are defined as the taxonomically identifiable skeletal remains from recent and past generations that are present in the surficial mixed layer on landscapes and seafloors [1,2]. Studies about ecology of death assemblages provide historical baselines (e.g., [3]) but also insights about modern spatial patterns (e.g., [4]). Two important taphonomic processes influencing the diversity and ecological structure of death assemblages are the time averaging and the postmortem lateral transport of shells. Time averaging is the co-occurrence of skeletal remains of individuals that died in different times reducing the temporal resolution of the study [2]. However, a positive point is that time averaging allows an accurate assessment of the species richness and displays a better mapping of ecological structure compared to any single census of the living assemblages [5]. Regarding to the lateral transport of skeletal remains in bottom sublittoral environments, it is rather small because [5]: lateral movement of shells is mainly within-habitat, most allochthonous shells are from immediately adjacent habitats, and drift transport affects particularly small thin shells.

Mollusca is the 2nd most diverse phylum of metazoans [6] with about 133000 extant described species [7], although many more species are likely still undiscovered. Most mollusks bear a shell of calcium carbonate that remains in the sediments longer after the organism's death providing a rich source of information about diversity [4,5]. Especially in marine ecosystems, mollusk death assemblages are a key source of historical data since the analyses of shells provides knowledge about structure (e.g., evenness, composition) and functional features (e.g., productivity, bioturbation potential) [1,8–11]. Therefore, mollusk death assemblages constitute a powerful and information-rich tool for analyzing diversity and setting biological baselines to gauge the effects of potential impacts (e.g., oil spill, eutrophication). In addition, studies about mollusk death assemblages are particularly relevant in those regions where knowledge is scarce, like the Western Tropical Atlantic.

Within the last ten years has occurred an accelerate development of approaches for measuring diversity [12–14]. Magurran [14] highlights the power of Robert Whittaker's approach that uses complementary facets of diversity (i.e., richness and compositional change) and levels (α-, β-, and γ-diversities). This approach extends to the analysis of taxonomic diversity, but also includes phylogenetic and functional diversities [14]. Phylogenetic diversity adds an evolutionary perspective to the biodiversity patterns [15] while functional diversity brings a potential connection with ecosystem functioning [16]. Therefore, the analysis of the three dimensions of diversity provide a holistic depiction of the biodiversity patterns and the underlying processes. For the analysis of compositional change (β-diversity *sensu lato*), Anderson et al. [17] proposed the use of turnover and variation for directional and non-directional compositional changes, respectively. Latter authors also highlight the power of multivariate measures for β-diversity assessment [17]. The partition of β-diversity patterns has been a controversial issue based on ecological and mathematical grounds [18]: An approach splits β-diversity in replacement and richness differences [19–21]; while another approach use replacement and nestedness [22–24]. Evenness also constitutes a key metric of the community when changes in species relative abundance are of interest [13]. Recently, other two unified frameworks for the analyses of diversity that include the relative abundance have been published: Pavoine et al. [12] proposed three types of overall measures (richness, skewness, and divergence) and Chao et al. [25] proposed Hill's numbers for measuring richness and differentiation. Given the bewildering array of approaches for describing diversity, we have chosen some of the above methods that better fit our objectives and type of data.

Habitat type is a tenet of benthic ecology affecting community structure and function [26,27]. Habitat type is a main driver of the diversity and assemblage structure of living

mollusks at local and regional scales [28]. Therefore, we use the habitat type as an organizing factor for studying mollusk death assemblages in the western Cuban shelf. Specifically, we focus on coral reefs and seagrass meadows, two iconic habitats of tropical shelves harboring a large diversity and complex communities [29,30]. Coral reefs occur over an important extension of Cuban Archipelago and display a large spatial variability regarding cover, average coral size, and health [31]. Given that most coral reef substrate is hard bottom, we selected the sandy spots located in the spur-and-groove biotope within the reef (hereafter "reef sands"). Seagrass meadows also occupy a broad extension of soft bottoms in Cuban shelf contributing to the sustaining of biodiversity and carbon storage [32].

In Cuban Archipelago, there is a large gap of knowledge in reference to the diversity of marine mollusk assemblages. There are several taxonomic publications describing new species of marine mollusks (mostly micro-mollusks); meanwhile only a few publications refer to the diversity and assemblage structure. In the Gulf of Batabanó, arguably the best-studied inner sea of the archipelago, only three studies on diversity patterns of living mollusks have been done [28,33,34]. Only one study exists about the deep-sea mollusk assemblages (about 1500 m depth) in the northwestern slope, reporting a very high diversity and potential transport of shells from the shelf [11]. Another three studies using sediment dating with $^{210}$Pb and $^{14}$C reported historical changes in mollusk death assemblages as response to human disturbances [35–37]. All the previous studies (with exception of [34]) only addressed taxonomic diversity, therefore phylogenetic and functional diversity patterns are unknown. Previous dead shells studies in Cuba (i.e., [35–37]) did not explicitly examine the effects of the spatial scales neither compare different types of habitats. Therefore, it is appropriate to increase the sampling coverage of mollusk death assemblages and to include additional dimensions to the taxonomic diversity (i.e., phylogenetic and functional).

This study is part of a large project aimed to compare the benthic biology between inner seas (gulfs) in the Western Cuban Archipelago, that encompass the Northwestern Caribbean Sea and the Southeastern Gulf of Mexico. The goal of our study is to document the taxonomic, phylogenetic, and functional diversity of mollusk death assemblages in sediments within two types of habitats and at two spatial scales. As part of the diversity assessment, we analyze the two components of diversity (richness and β-diversity) using the species, phylogenetic, and functional dimensions. We also address the evenness and taxonomic/functional composition of death assemblages by having in account the relative abundance of the species and functional groups, respectively.

## Materials and methods

### Study regions

We sampled two shallow seas (hereafter gulfs) in the western part of Cuban Archipelago (Fig 1): Gulf of Batabanó (GB) in the Northwestern Caribbean Sea and Gulf of Guanahacabibes (GG) in the Southeastern Gulf of Mexico. Both gulfs have dominance of sedimentary bottoms, with shoreline covered by mangrove forests, extensive areas of seagrass meadows (mostly *Thalassia testudinum*) in the central part and bordered by coral reef tracts in the shelf border [35,38]. Reef sand sites in GB and GG are within the coral reef systems Canarreos and Los Colorados, respectively. Canarreos is a reasonably healthy reef system with the largest individual coral size and reef complexity of the whole Cuban Archipelago [31]. Meanwhile, Los Colorados contains a worrying number of depleted reefs with low living coral cover [31]. Seagrass sites differed in the proximity of the shelf border and therefore in the influence of open waters. GB sites, in the shelf border, are subjected to an intense physical disturbance by waving and currents coming from the Caribbean Sea and are relatively far from the pollution sources

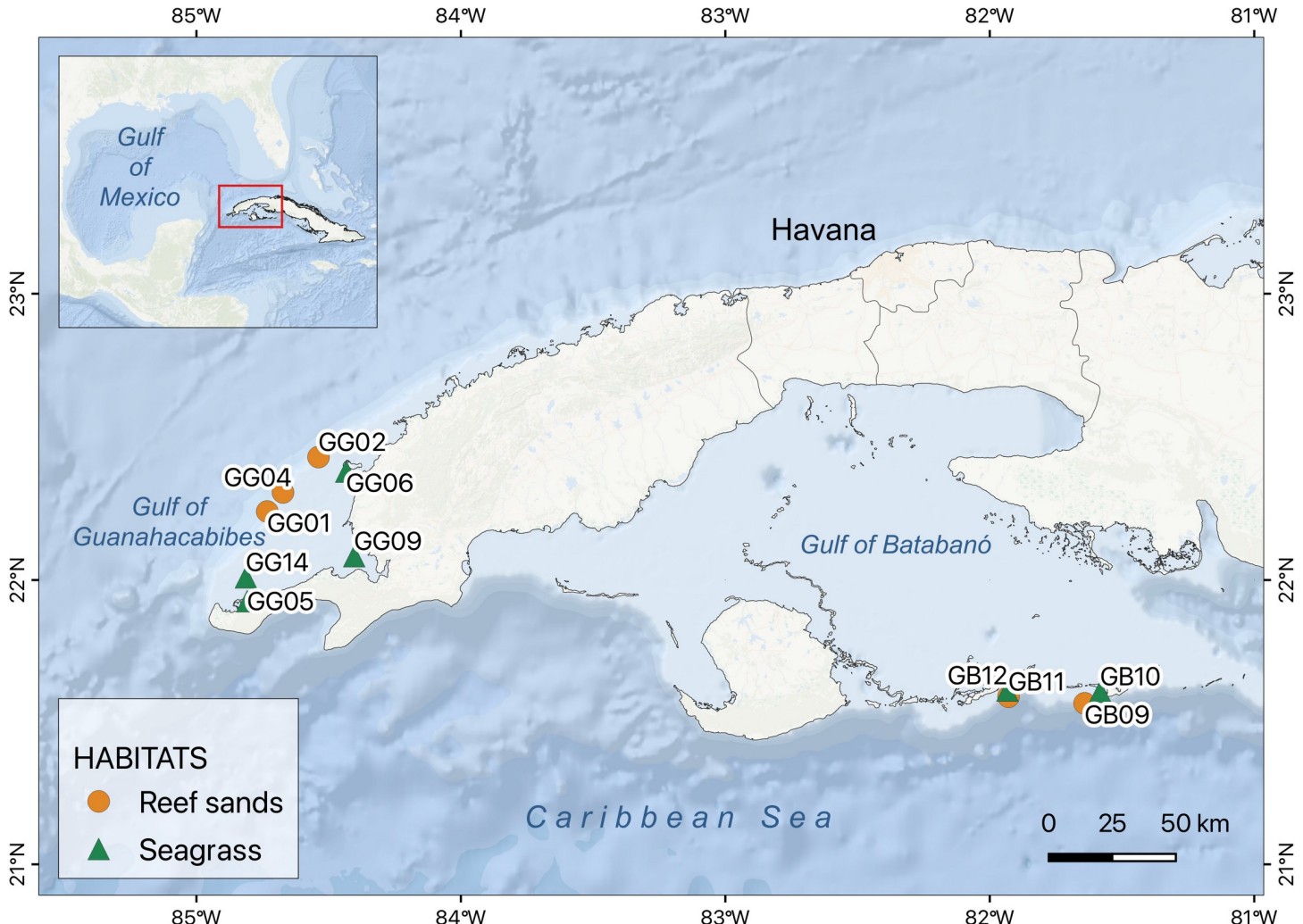

**Fig 1. Map of the study region.** Sampling sites within the gulfs of Batabanó (GB, 4 sites) and Guanahacabibes (GG, 7 sites). Reef sands (orange circles) and seagrass meadows (green triangles). Spatial data from GADM v3.6.

in the Cuba Island. Meanwhile GG sites are closer to land, with slower flushing of water by hydrodynamics and more exposed to anthropogenic disturbances such as eutrophication and turbidity [37].

Other similarities between GB and GG are the average depth (about 6–8 m) and the hydrological variables: temperature 27–29°C, salinity 35.1–37.4 PSU, dissolved oxygen 6.1–8.0 mg L$^{-1}$, and pH 7.9–8.2. However, there are main differences between GB and GG, such as maximum depth (12 versus 28 m, respectively) and extension (20000 versus 1900 km$^2$).

## Sampling and sample processing

We sampled 11 soft-bottom sites in two expeditions to GB (November 1$^{st}$–5$^{th}$, 2015) and GG (May 30$^{th}$–June 10$^{th}$, 2014) (Table 1). We selected two types of habitats broadly distributed across the gulfs: coral reefs and seagrass meadows. In coral reefs, we sampled in sandy spots within spur-and-groove structures in forereefs. Therefore, our work focuses on mollusk

**Table 1. Characterization of the sampling sites in the gulfs of Batabanó and Guanahacabibes.**

| Gulf | Habitat | Site | Latitude (N) | Longitude (W) | Depth (m) | S+C (%) | TOM (%) | Reps. | N | S | Observations |
|------|---------|------|--------------|---------------|-----------|---------|---------|-------|------|-----|--------------|
| GB | Reef sands | GB09 | 21˚34.041 | 81˚38.484 | 30 | 18 | 8 | 3 | 1553 | 181 | "Acuario" buoy. Outer reef, clean waters, spur-and-groove biotope, fine and white sands |
| | Seagrass | GB10 | 21˚36.527 | 81˚34.959 | 3 | 26 | 10 | 3 | 744 | 79 | Close to entrance channel of Cayo Largo marina. Meadow (high density) of *T. testudinum* and *S. filiforme* |
| | Reef sands | GB11 | 21˚35.427 | 81˚55.764 | 14 | 12 | 9 | 3 | 1118 | 166 | Outer reef, clean waters, spur-and-groove biotope, fine and white sands |
| | Seagrass | GB12 | 21˚36.744 | 81˚56.016 | 4 | 55 | 9 | 3 | 355 | 55 | Meadow (high density) of *T. testudinum* and algae |
| GG | Reef sands | GG01 | 22˚14.397 | 84˚43.933 | 20 | - | 8 | 3 | 364 | 83 | Outer reef, clean waters, spur-and-groove biotope, many gorgonians and sponges, scarce sediment made of coarse and white sands |
| | Reef sands | GG02 | 22˚25.818 | 84˚32.303 | 18 | 8 | 9 | 1 | 139 | 40 | Outer reef, clean waters, spur-and-groove biotope, many gorgonians and sponges, scarce sediments, signs of strong hydrodynamics |
| | Reef sands | GG04 | 22˚18.404 | 84˚40.360 | 18 | 13 | 10 | 3 | 660 | 142 | Outer reef, spur-and-groove biotope, many gorgonians and corals, more areas of sandy sediments than in previous sites |
| | Seagrass | GG05 | 21˚55.556 | 84˚48.483 | 4 | 43 | 13 | 3 | 752 | 83 | Meadow composed mostly of *T. testudinum* (medium density) and many algae. Sediment gray and with white sand grains, darker in the top10 cm |
| | Seagrass | GG06 | 22˚22.792 | 84˚26.019 | 5 | 61 | 10 | 3 | 576 | 58 | Meadow composed mostly of *T. testudinum* (medium density), many algae, muddy-sandy bottom rich in detritus, and many epiphytes on blades. Sediment was light gray along the entire core. Some shells >5 cm |
| | Seagrass | GG09 | 22˚05.030 | 84˚24.206 | 5 | 36 | 8 | 2 | 474 | 71 | Meadow of *T. testudinum* (high density), muddy-sandy bottom with many invertebrates |
| | Seagrass | GG14 | 22˚00.582 | 84˚48.783 | 5 | 6 | 5 | 2 | 378 | 68 | Meadow of *T. testudinum* (mid-low density) with short blades and many green algae |
| Total | | 11 | | | | | | 29 | 7113 | 393 | |

Abbreviations: GB = Gulf of Batabanó, GG = Gulf of Guanahacabibes, S+C = silt plus clay fractions, TOM = total organic matter, Reps. = number of replicates, N = abundance of dead individuals, S = number of species.

assemblages living in sediments and not on those assemblages living in nearby hard substrates of the reef.

Sediment samples were collected by SCUBA divers using a box core (= sampling unit) with an effective sampling area of 100 cm$^2$. Three sampling units were collected per site, but in three sites we missed some replicates because diving-related issues (Table 1). The box core was pushed 10 cm into the bottom collecting approximately 1000 cm$^3$ of sediments. A potential limitation with this sampling is that it may miss deep burrowing species inhabiting below 10 cm deep. Sediments were sieved onboard with filtered seawater through a 500-μm sieve and preserved in 70% ethanol. We chose this mesh size to capture the micro-mollusks [39]. In the laboratory, all the mollusk shells were sorted and counted under a stereomicroscope SZX7. In the case of the bivalves, each disarticulated valve was counted as an individual. The dead individuals were separated from the living ones, and the complete shells were kept for analysis. Large skeletal fragments retaining the apex or more than half of the hinge line were also counted as dead individuals. Mollusks were identified to species level using taxonomic literature [40–43].

## Phylogeny and biological traits

Phylogeny followed the Linnean system of classification (i.e., class, order, family, genus, and species) since no other data were available for most of the species (e.g., DNA-based phylogeny). The taxonomy of the species was obtained by the Match taxa tool of the World Register of Marine Species using only the accepted names [7].

We classified the species after five biological traits: (i) life mode, (ii) feeding type, (iii) relationship with the substrate, (iv) mobility, and (v) shell attachment (S1 Table). The biological traits were mainly obtained from the database Neogene Marine Biota of Tropical America (NMITA) [44], including support literature for mollusk ecology [40,41,45,46]. All categories within traits were adapted following the codes given at NMITA. The biological trait information was usually given at genus or family levels, so we extrapolated at the species level.

Gastropods were classified only after their life mode and the feeding type as no other information was available at NMITA or in the consulted literature. For the Scaphopods, we were able to fill in their traits after Tunnell et al. [42].

## Data analysis

We measured the diversity using two complementary components [14]: richness (α- and γ-diversity) and β-diversity. We analyzed the three dimensions of diversity: species, phylogenetic, and functional diversity. We also analyzed the ecological structure of the assemblage considering the relative abundance of the species using dominance curves, heat maps, and numerical ordination according to the appropriate scale. The statistical design included two spatial scales: regional that encompassed the gulf and local that referred to the sampling site. We also included habitat in the design because it has a significant impact on the diversity in the studied region [28]. In general, the criterion for testing differences between groups was the overlapping of the 0.95 CIs calculated on basis of permutations.

**Richness.**   For the species richness, we built accumulation curves versus individuals using the effective number of species (Hill's number 0) [47]. The accumulation curves included the observed richness, the asymptotic estimates (only at regional scale), and 0.95 confidence intervals (CIs) generated by 500 permutations using the functions *inext* and *gginext* in the R package iNEXT 3.0.0 [48]. In the second step, we calculated the expected richness at the same level of abundance using rarefaction.

For the phylogenetic richness, we used the Faith's phylogenetic diversity (PD sensu Faith [49]), which is the sum of the branch lengths in a phylogenetic tree. PD measures the amount of evolutionary history across species [15]. The phylogenetic tree was based on the Linnean hierarchy and built with the function *linnean* in the package BAT 2.9.2 [50].

For the functional richness, we used the Petchey and Gaston's functional diversity (FD sensu Petchey and Gaston [51]), which is the sum of the branch lengths in a functional tree. FD measures the overall distance between the different functional units (i.e., species) of a given assemblage [52]. Related with functional diversity, we used response traits that focus on the organism's response to its environment and does not imply that a trait necessarily influences an ecological process [53]. The functional tree was built in the package BAT with the function *tree.build* using the following settings: the traits were converted from categorical to numerical (i.e., dummy variables), Gower distance as resemblance measure, and modified Neighbor-Joining algorithm as linkage strategy. Species, phylogenetic, and functional richness were calculated using the function *alpha* in the R package BAT with rarefied samples (i.e., the same sample size) and using 1000 permutations for generating 0.95 CIs.

**β-diversity.**   We followed the framework proposed by Cardoso et al. [21,54] that analyzes species-, phylogeny-, and functional-based β-diversity. These three dimensions of total β-diversity were in turn decomposed into two partitions [19,20]: replacement and richness difference. For the species-based dimension, we built β-diversity accumulation curves for evaluating the effects of sampling completeness [55]. The pairwise values of dissimilarity were averaged over the relevant scale/level. That is, at regional scale all pairwise inter-site dissimilarities within a gulf were averaged (e.g., for GB: GB09 versus GB10, GB09 versus GB11, etc.). For

habitats, pairwise inter-site dissimilarities in a habitat nested within gulf were used (e.g., for reef sands in GB: GB09 vs. GB11; for seagrass in GB: GB10 vs. GB12). We built β-diversity accumulation curves and calculated β-diversity using the function *beta.accum* and *beta*, respectively, in the R package BAT. The phylogenetic and functional trees were the same as for richness analyses. The measure of dissimilarity was the Sorensen index that uses incidence data since changes in composition between samples was the target of the analysis. Calculations were made on rarefied samples (at level of smallest sample size) to reduce the effects of differences in abundance and 100 permutations were done to generate 0.95 CIs around the mean β-diversity.

**Evenness.** This property of the assemblage structure was represented in curves of dominance of species rank (x-axis in log scale) versus relative abundance [56]. We also quantified the rareness based on the number of species with one individual (i.e., singletons) and two individuals (i.e., doubletons).

**Taxonomic composition.** We used the software PRIMER v7 for describing the assemblage structure [56]. We applied the function *simper* to identify those species that most contribute to the 70% of similarity (Bray-Curtis index) for a particular scale/level (i.e., gulf, habitat, or site); such species can be considered as the typical or characteristic of that scale/level [56]. For this analysis, we averaged the abundance of species over the sites. To visualize the typical species, we built heat maps of their relative abundance. The typical species in the heat maps were ordered based on their similarity across the samples using an index of association [56]. In the case of analyses at site scale, instead of a heat map, we preferred to build a numerical ordination by non-metric multidimensional scaling (NMDS) with all the replicates to visualize the similarity pattern across sites. NMDS was built on basis of a similarity matrix calculated with the Bray-Curtis index, no transformation of data was done. A permutational analysis of variance (PERMANOVA) [57] was made for testing differences in the multivariate assemblage structure following the nested statistical design mentioned above: GULF as main fixed factor, HABITAT as random factor nested within GULF, and SITE as random factor nested within HABITAT. The settings of PERMANOVA were Bray-Curtis as measure of resemblance, 9999 permutations for building the null distribution, sum of squares type III (partial), and permutation of residuals under a reduced model.

**Functional composition.** We built four matrices of trait × samples by summing species based on their biological traits: feeding type, relationship organism-substrate, mobility, and type of attachment of the shell to substrate (hereafter shell attachment). The trait life mode (i.e., benthic or holoplanktonic) was not included because there were only six holoplanktonic species. The samples were aggregated according to the scale/level of the analyses (i.e., gulf, habitat, or site). We represented the relative contribution of each trait category to the total abundance in stacked bar plots. Categories within a trait were not independent (i.e., they summed 100%), therefore no statistical tests were done to compare them. For the functional composition across sites, we ordered them according to their similarity based on species abundance using the Bray-Curtis index.

All graphs were made in the package Tidyverse [58] in R.

## Results

### General features

We identified 7113 dead individuals belonging to 393 species, 242 genera, 101 families, 20 orders, and three classes. Holoplanktonic mollusks occurred only in GB and were represented by six species and 41 individuals (0.6% of total abundance). The full matrix of mollusk species × sites was given in the S2 Table. The most abundant species were *Parvilucina* spp.

(8%), *Eulithidium* sp. (6%), *Cerithium litteratum* (4%), *Barbatia domingensis* (4%), *Eulithidium bellum* (3%), *Cerithium eburneum* (3%) and *Alvania auberiana* (3%).

### Regional scale (Gulfs)

**Richness.**    We identified 280 and 265 mollusk species in the gulfs of Batabanó and Guanahacabibes, respectively. The two accumulation curves of species did not reach asymptotes (S1 Fig). However, the sampling coverage was 0.98 for both gulfs, suggesting a high completeness of the sampling effort. At a level of rarefaction of 3000 individuals, no significant differences of species richness occurred between gulfs (Fig 2A). Nevertheless, the phylogenetic and

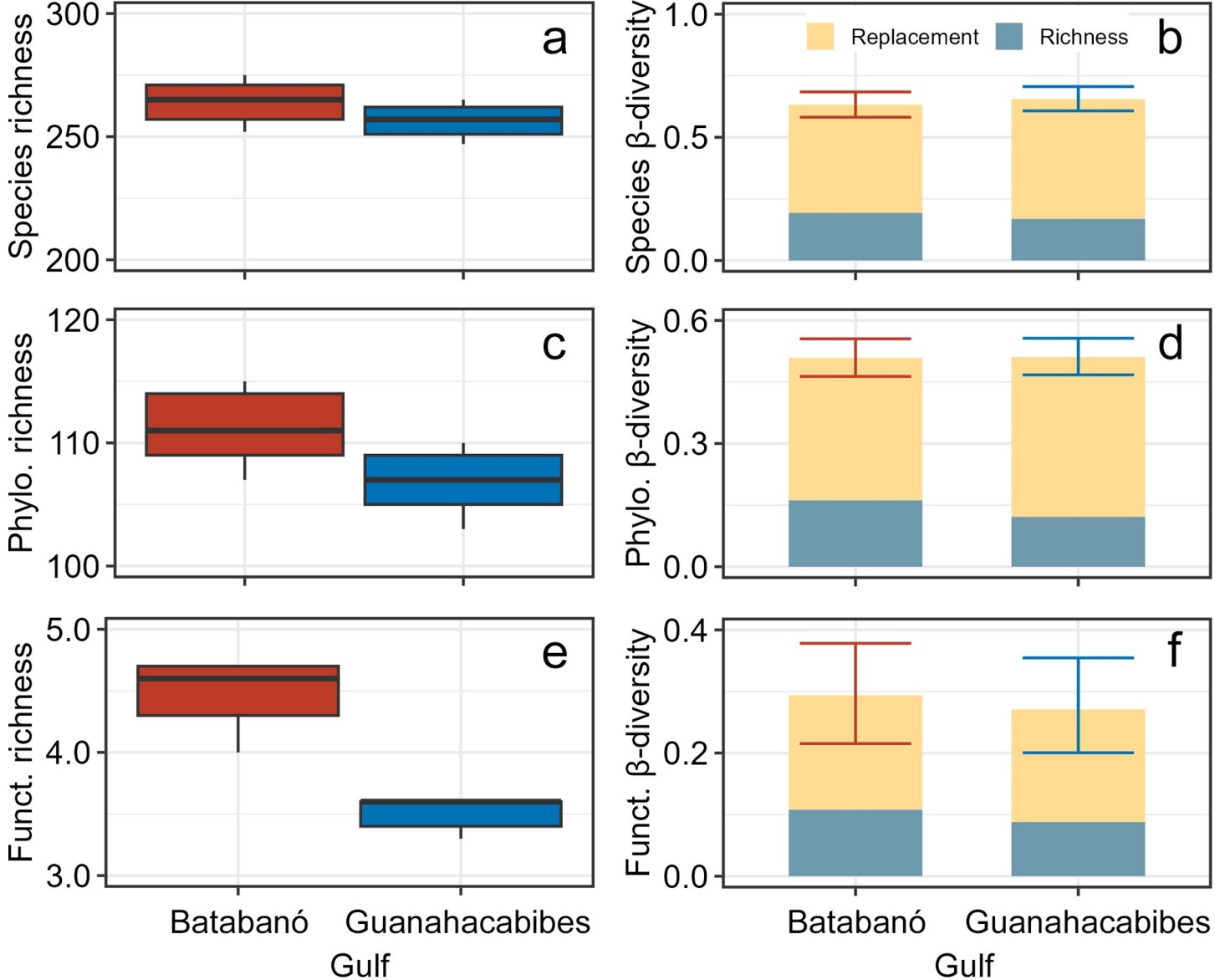

**Fig 2. Regional richness and mean β-diversity of mollusk death assemblages in the gulfs of Batabanó and Guanahacabibes.** (a) Species richness. (b) Species β-diversity. (c) Phylogenetic richness. (d) Phylogenetic β-diversity. (e) Functional richness. (f) Functional β-diversity. Samples were rarefied to 3000 individuals in (a), (c), and (e); while in (b), (d), and (f) were rarefied to the lowest number of individuals between pairs of sites. Two additive partitions of β-diversity are shown: replacement and richness difference. Horizontal lines = mean, boxes (or whiskers) = 0.95 confidence intervals, and vertical lines = range. Phylo. = phylogenetic, Funct. = functional.

functional richness were different between gulfs as indicated by the non-overlapped CIs (Fig 2C and 2E). In general, GB tended to have larger richness than GG.

**β-diversity.** The accumulation curves suggested a modest decrease of mean β-diversity with the increase of the replication suggesting that the current sample size (n = 3, for most of the sites) is enough for a robust comparison (S1 Fig). The total species β-diversity was high ($> 0.6$) and similar for GB and GG (Fig 2B). The replacement partition was higher than richness difference partition in both gulfs. Phylogenetic β-diversity was similar between the gulfs (around 0.5), and with replacement partition being higher than richness difference as well (Fig 2D). Functional β-diversity had the same pattern, although total values of β-diversity were lower ($< 0.5$) (Fig 2F).

**Evenness.** The evenness was similar in the two gulfs as indicated by the similar shape of the dominance curves (Fig 3A). The main difference between the two curves is the level of dominance of the most abundant species, which was 17% and 7% in GG and GB, respectively. Nine species contributed less than 5% each to the total abundance in both gulfs followed by a long tail of rare species represented by singletons (71 in GB and 77 in GG) and doubletons (38 in GB and 48 in GG).

**Taxonomic composition.** The heat map representing the assemblage structure of 25 typical species for both gulfs suggested that they were largely the same, despite some variability in the relative abundances (S2 Fig). In other words, only two species (*Smaragdia viridis* and *Ervilia nitens*) occurred in a single gulf (GB), the other 23 species occurred with variable abundance in both gulfs.

**Functional composition.** The trait feeding type was represented by five dominant categories, the category "others" lumped together deposit feeders, herbivores/suspension feeders, and herbivores/carnivores. Herbivores dominated the functional trophic structure (Fig 4A). The most striking difference between gulfs was the larger abundance of chemosynthetic deposit feeders in GG. The trait relationship organism-substrate included three dominant categories, and the category "others" lumped together nestler, borer, and the mixed categories epifaunal/ semi-infaunal/borer. The dominant category was infauna, followed by epifauna, and facultative epifauna/infauna (Fig 4B). The two gulfs had a similar contribution of each category to the total abundance. The trait mobility was represented by two dominant categories, and the category "others" lumped together swimming, sessile, and the mixed categories actively mobile/ sedentary/sessile. The actively mobile forms strongly dominated in the two gulfs in terms of abundance, followed by sedentary forms (Fig 4C). The trait shell attachment included two dominant categories and the category "others" lumped together cemented and mixed categories unattached/bysally attached. The unattached forms were dominant in both gulfs, however the bysally attached forms were more important in GB when compared to GG (Fig 4D).

## Habitat

**Richness.** We used the nested structure to compare two habitat types within the corresponding gulfs. The species accumulation curves were not asymptotic (S1 Fig). However, the sampling coverage was high as well, ranging between 0.95 and 0.98. Reef sands had higher mollusk species richness than seagrass; and within seagrass meadows, GG had higher richness than GB (Fig 5a). Phylogenetic richness had the same pattern, with higher richness in the reef sands, intermediate in seagrass from GG, and lowest in seagrass from GB (Fig 5C). Functional diversity showed a different pattern, with higher richness in reef sands from GB, and lower in the other three groups (Fig 5E).

**β-diversity.** β-diversity accumulation curves showed a moderate decrease from 1 to 3 sampling units (S1 Fig) indicating that current sample size was enough for a robust

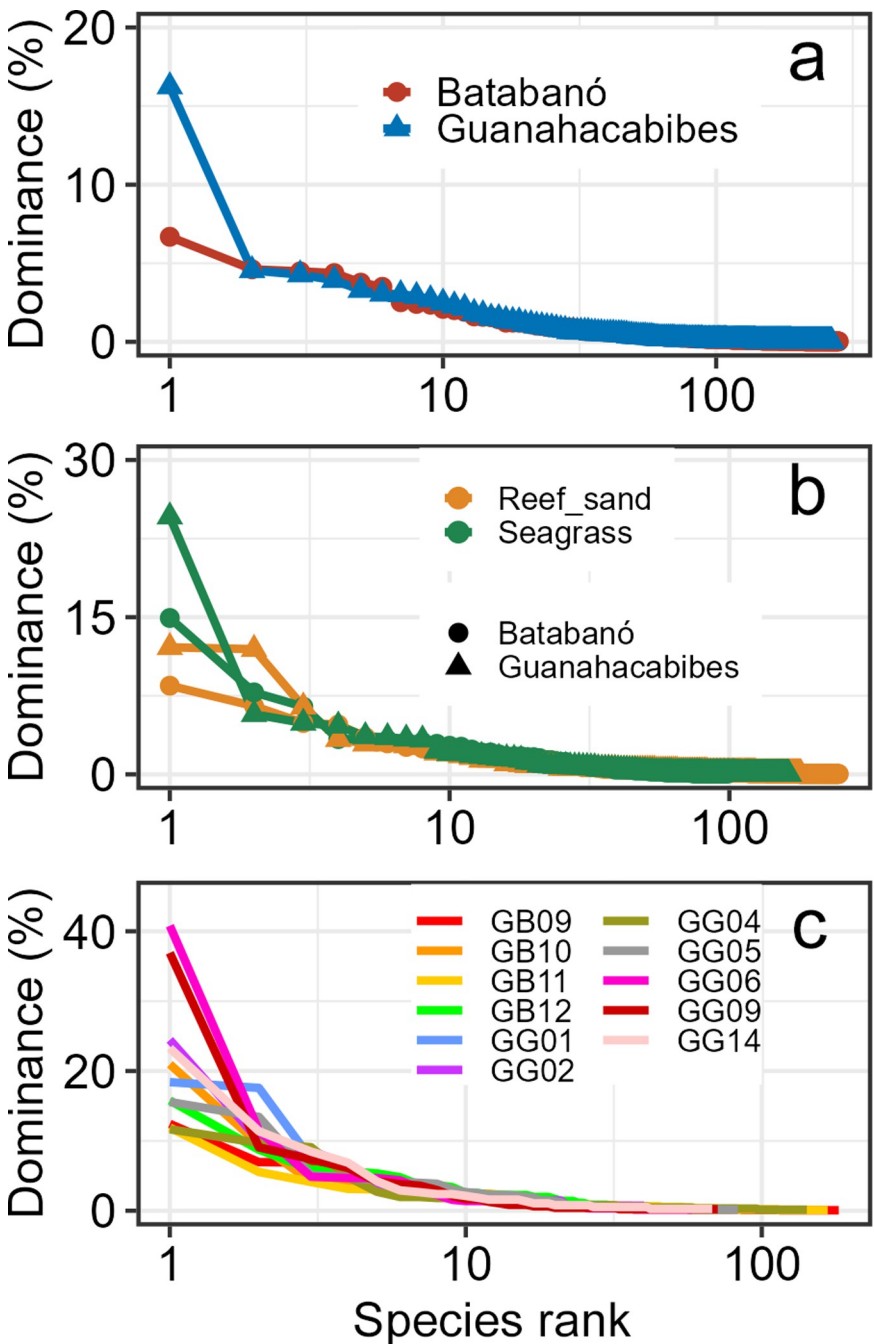

**Fig 3. Evenness of mollusk death assemblages.** Dominance curves of relative contribution of species to the total abundance versus abundance species rank. (a) Regional scale. (b) Habitat. (c) Site scale.

comparison. There were no differences of species β-diversity between reef sands and seagrass in GB. However, in GG the species β-diversity was slightly higher in seagrass compared to reef sands (Fig 5B). The replacement partition was much more important than richness differences for all the combinations habitat-gulf. The phylogenetic β-diversity was similar between the two habitats nested within gulfs, and the replacement partition was also the larger (Fig 5D). The functional β-diversity did not show differences between habitats nested within gulfs (Fig 5F).

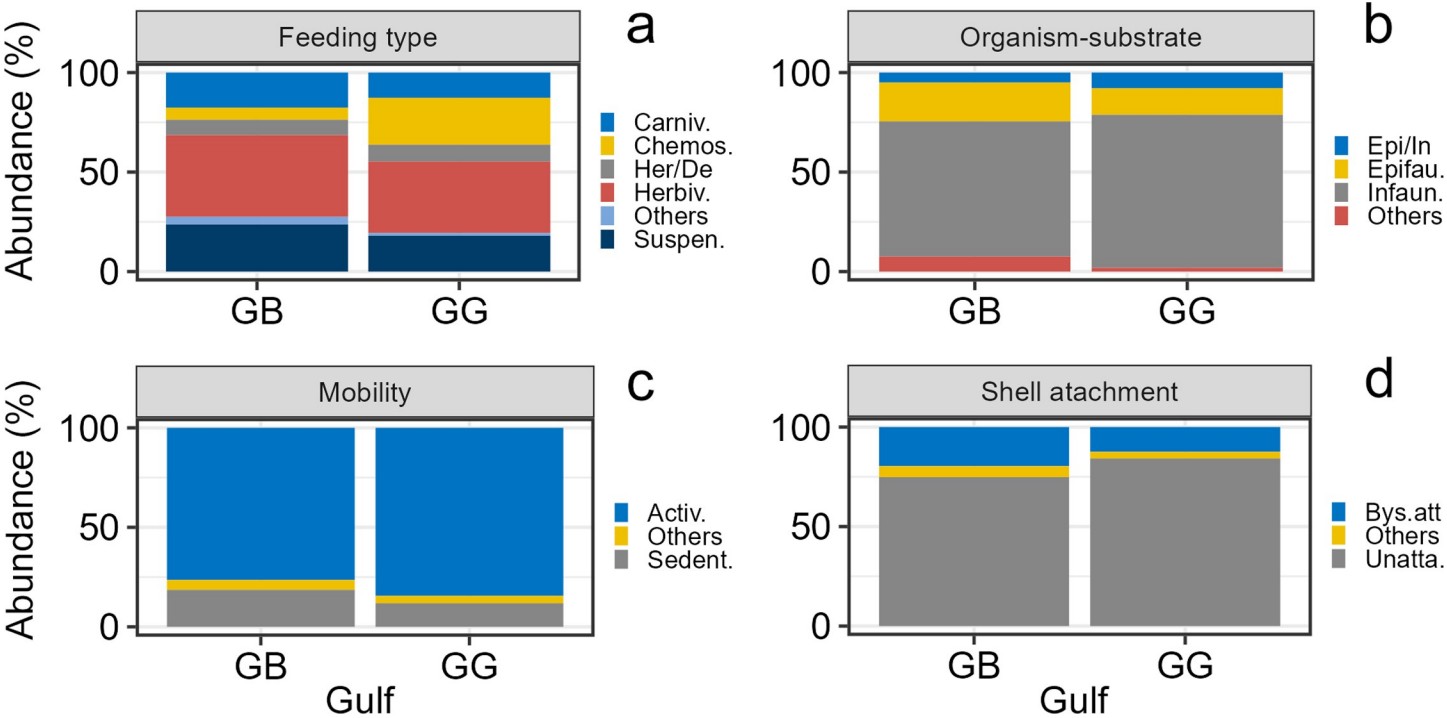

**Fig 4. Regional functional composition of mollusk death assemblages in the gulfs of Batabanó (GB) and Guanahacabibes (GG).** (a) Feeding types, including carnivore, chemosymbiotic deposit feeder, herbivore/deposit feeder (facultative), herbivore, suspension feeder, and others. (b) Relationships organism–substrate, including infauna, epifauna, epifauna/infauna (facultative), and others. (c) Mobility, including actively mobile, sedentary, and others. (d) Shell attachment, including unattached, bysally attached, and others. The traits in (b), (c), and (d) were calculated only for bivalves. See the text for details about "others".

But the richness differentiation partition increased relatively (around 50% of total β-diversity) suggesting that loss/gain of trait categories played a more important role than simple replacement of one category by another.

**Evenness.** There were differences among the habitats nested within gulfs as indicated by the four curves (Fig 3B). However, seagrass had larger dominance compared to reef sands, given by the two most abundant species (23% and 15% in GG and GB, respectively). The 2nd and 3rd most abundant species for both types of habitats contributed between 5% and 12% of the total, suggesting a rather high evenness. As expected, there was a long tail of rare species represented by an average (over the four combinations) of 52 singletons and 31 doubletons.

**Taxonomic composition.** The heat map of typical species suggested sharp differences between habitats. In GB, there were 14 species typical of reef sands but absent in seagrass, and conversely three species typical of seagrass but absent in reef sands (Fig 6). In GG, there were eight species typical of reef sands but absent in seagrass, and four species typical of seagrass but absent in reef sands.

**Functional composition.** Feeding type structure had notable differences between habitats (Fig 7A). The feeding type structure in reef sands from the two gulfs was similar, with dominance of herbivores, followed by carnivores and suspension feeders. Feeding type structure in seagrass was different compared to reef sands, but also had differences between GB and GG. For instance, in GB suspension feeders was the most important category, while in GG was chemosynthetic deposit feeders. The relationship organism-substrate was quite different between habitats nested within gulfs (Fig 7B). In reef sands, epifauna represented more than half of the abundance, while in seagrass they were rather scarce. Seagrass, in both gulfs, were largely

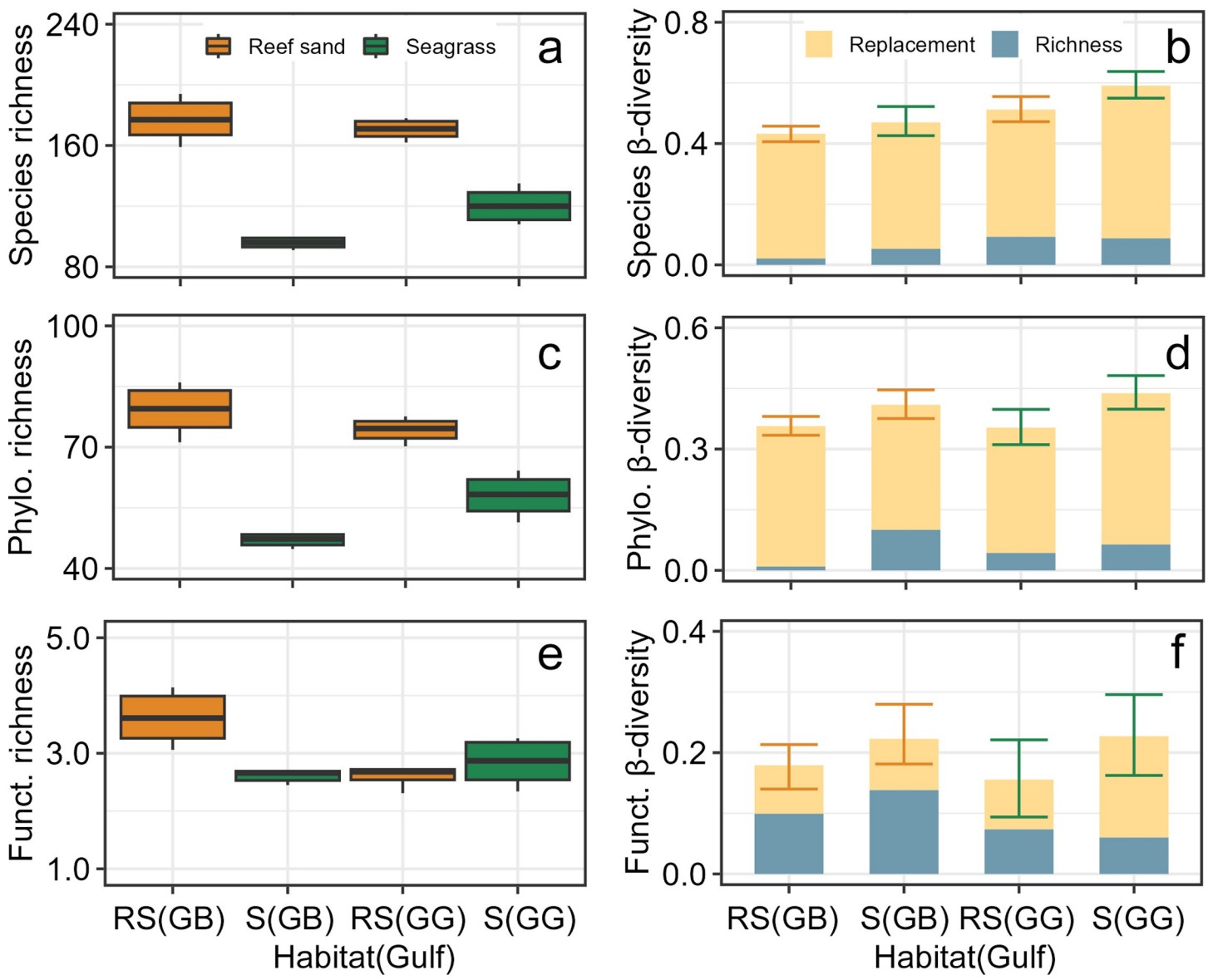

**Fig 5. Habitat richness and mean β-diversity of mollusk death assemblages in reef sands (RS) and seagrass (S) within the gulfs of Batabanó (GB) and Guanahacabibes (GG).** (a) Species richness. (b) Species β-diversity. (c) Phylogenetic richness. (d) Phylogenetic β-diversity. (e) Functional richness. (f) Functional β-diversity. Samples were rarefied to 1000 individuals in (a), (c), and (e); while in (b), (d), and (f) were rarefied to the lowest number of individuals between pairs of sites. Two additive partitions of β-diversity are shown: replacement and richness difference. Horizontal lines = mean, boxes (or whiskers) = 0.95 confidence intervals, and vertical lines = range. Phylo. = phylogenetic, Funct. = functional.

dominated by infauna. Mobility had a similar pattern with sharp differences between habitats (Fig 7C). In reef sands, sedentary forms were similarly abundant to actively mobile forms. However, in seagrass the sedentary forms virtually disappeared, and only actively mobile forms occurred. Shell attachment had a similar pattern to the other two previous traits. There were notably differences between habitats nested within gulfs (Fig 7D). In reef sands, byssally attached and unattached forms were roughly equally important, and the category "others" occurred in much lesser proportion. While in seagrass, unattached forms largely dominated the assemblages.

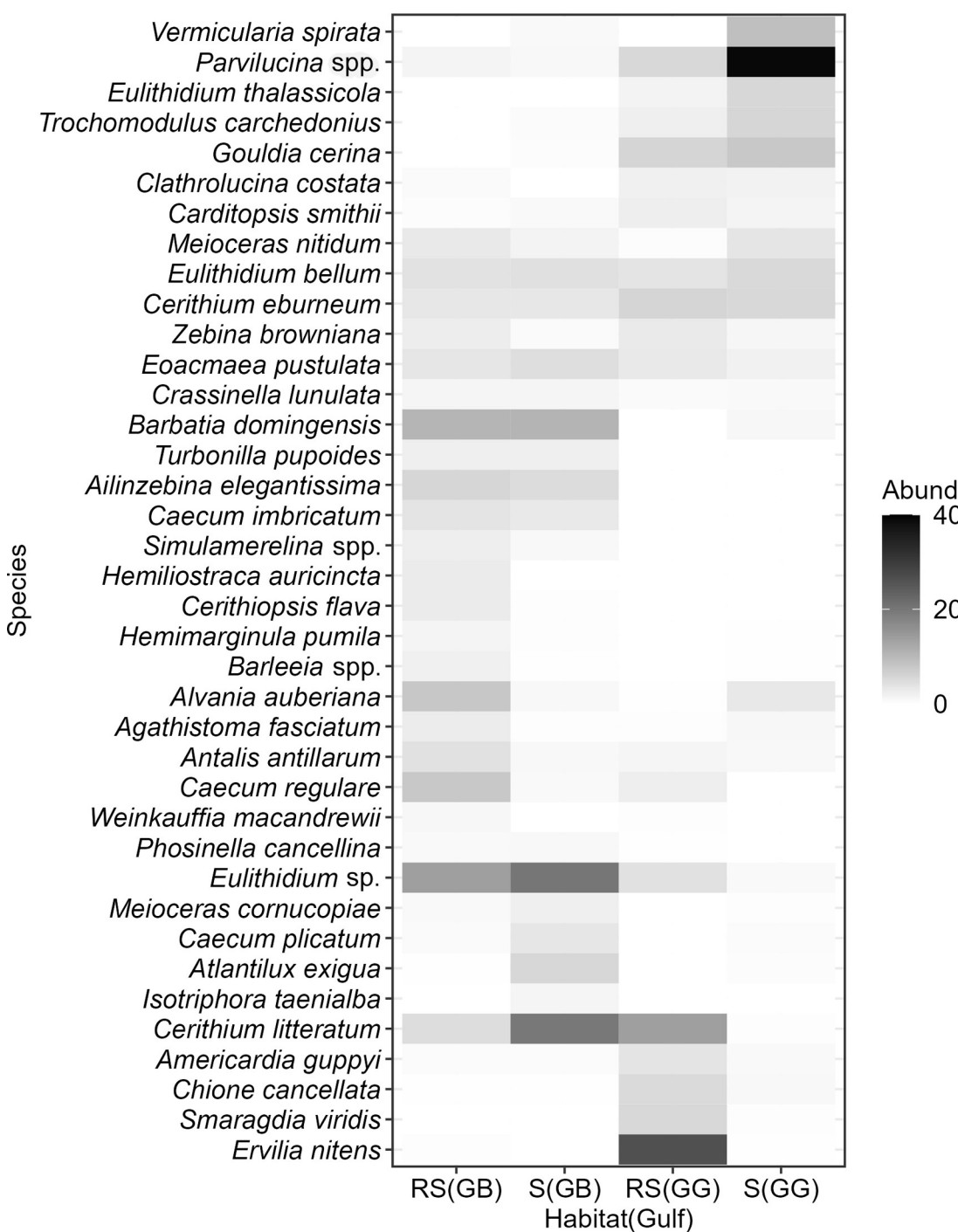

**Fig 6. Assemblage structure of mollusk death assemblages in reef sands (RS) and seagrass (S) within the gulfs of Batabanó (GB) and Guanahacabibes (GG).** Heat map of relative abundance (in %) of those typical species that contribute to the 70% of similarity within each habitat. Note that species were ordered by their similarity across the habitat-gulf combinations.

## Local scale (sites)

**Richness.** The shapes of the species accumulation curves for sites were non-asymptotic (S1 Fig). However, the sampling coverage was relatively high and ranged from 0.86 at GG02

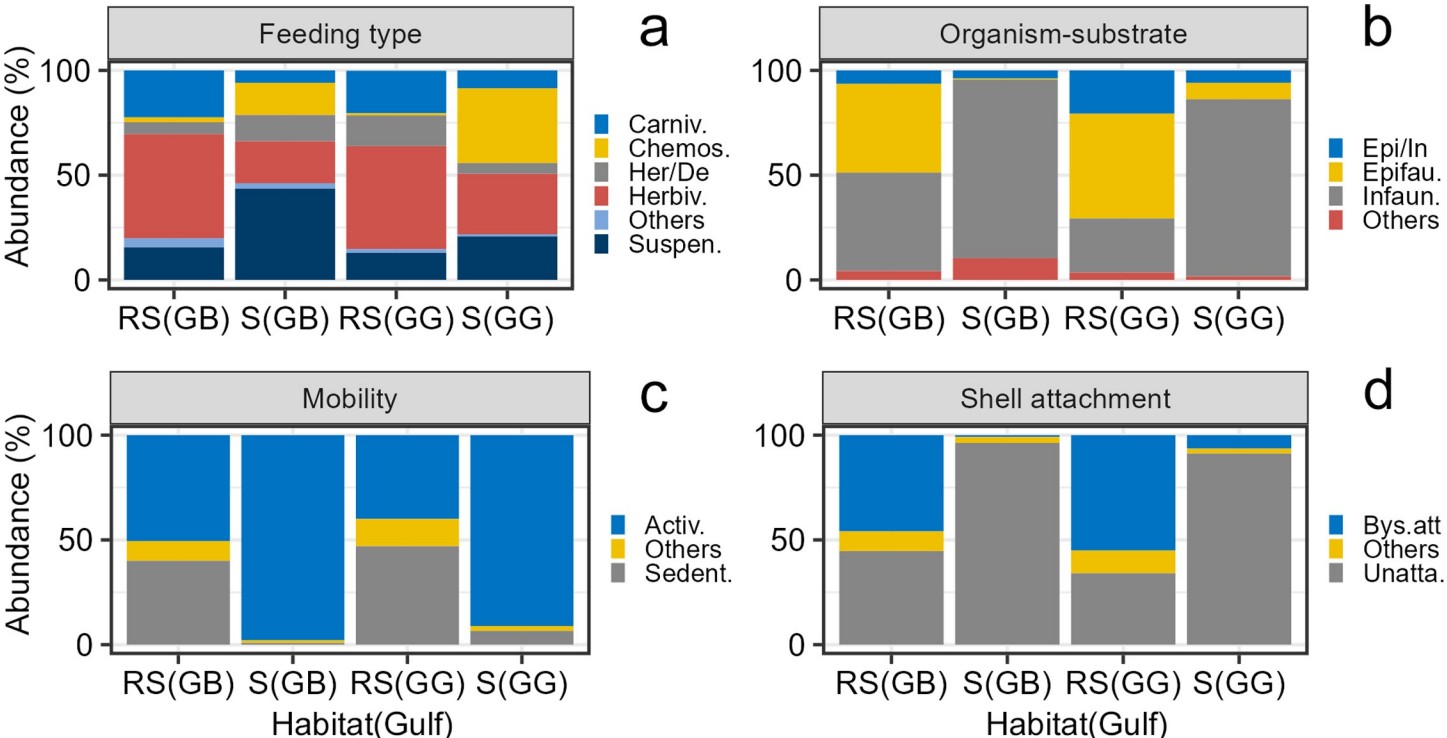

**Fig 7. Habitat functional composition of mollusk death assemblages in reef sands (RS) and seagrass (S) within the gulfs of Batabanó (GB) and Guanahacabibes (GG).** (a) Feeding types including carnivore, chemosymbiotic deposit feeder, herbivore/deposit feeder (facultative), herbivore, suspension feeder, and others. (b) Relationships organism - substrate including infauna, epifauna, epifauna/infauna (facultative), and others. (c) Mobility, including actively mobile, sedentary, and others. (d) Shell attachment, including unattached, bysally attached, and others. The traits in (b), (c), and (d) are only for bivalves. See the text for details about "others".

(with single replicate) to 0.97 at GB09 and GB10. The most striking point in the curves was the large abundance of dead shells in the reef sand sites GB09 and GB11. The species richness varied significantly across sites with the most obvious differences between reef sand sites (GB09, GB11, GG01, GG02, and GG04) with higher richness and seagrass sites with lower richness (Fig 8A). The phylogenetic richness showed a similar pattern with high variability across sites, and significant differences between reef sand and seagrass sites (Fig 8C). Functional richness did not show differences across sites due to the overlapping of the 0.95 CIs (Fig 8E).

**β-diversity.**   Species β-diversity was around 0.5, with no differences among sites as suggested by the overlapping of the 0.95 CIs. The most important contribution to the total β-diversity was given by species replacement (Fig 8B). Phylogenetic β-diversity was in general lower than 0.5, but again without clear differences among sites. Most of the phylogenetic β-diversity was given by species replacement (Fig 8D). Functional β-diversity had a rather similar pattern, without differences among sites, but a larger relative contribution of richness differences to the total β-diversity (Fig 8F).

**Evenness.**   The evenness was similar for the 11 sites based on the dominance curves (Fig 3C). However, two seagrass sites in GG had higher dominance of a single species (GG06 with 41% and GG09 with 37%). The long tail of rare species was represented as average (over the 11 sites) by 37 singletons (range 19–68) and 18 doubletons (range 4–38).

**Taxonomic composition.**   For analyzing changes in taxonomic composition at site scale, we searched for the similarity patterns across sites. A NMDS ordination plot of the 29 samples (= points) based on species abundance indicated a clear pattern based on the habitat type (Fig 9). Reef sands and seagrass sites clustered apart from each other in the plot and showed

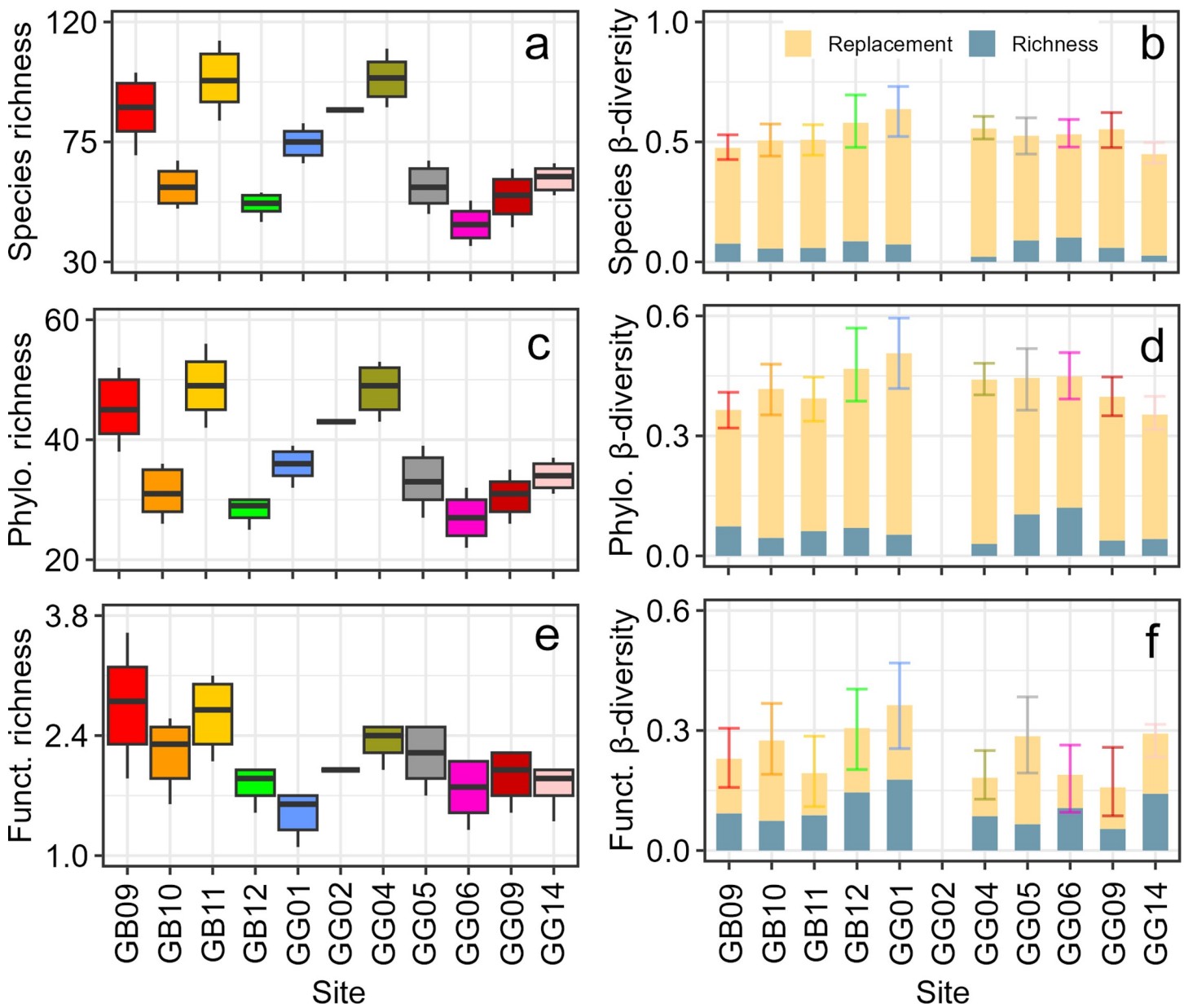

**Fig 8. Local richness and mean β-diversity of mollusk death assemblages in 11 sites within the gulfs of Batabanó (GB) and Guanahacabibes (GG).** (a) Species richness. (b) Species β-diversity. (c) Phylogenetic richness. (d) Phylogenetic β-diversity. (e) Functional richness. (f) Functional β-diversity. Samples were rarefied to 300 individuals in (a), (c), and (e); while in (b), (d), and (f) were rarefied to the lowest number of individuals between pairs of sites. Two additive partitions of β-diversity are shown: replacement and richness difference. Horizontal lines = mean, boxes (or whiskers) = 0.95 confidence intervals, and vertical lines = range. Phylo. = phylogenetic, Funct. = functional.

significant differences (PERMANOVA, S3 Table). However, differences in assemblage structure between gulfs were less marked and non-significant (Fig 9, PERMANOVA S3 Table). The variability between replicates from the same site was also considerable and statistically significant (PERMANOVA, S3 Table) suggesting high variability in the assemblage structure. Indeed, factors HABITAT and SITE explained about one third each of the total variance, meanwhile the remaining third was unexplained (= residual).

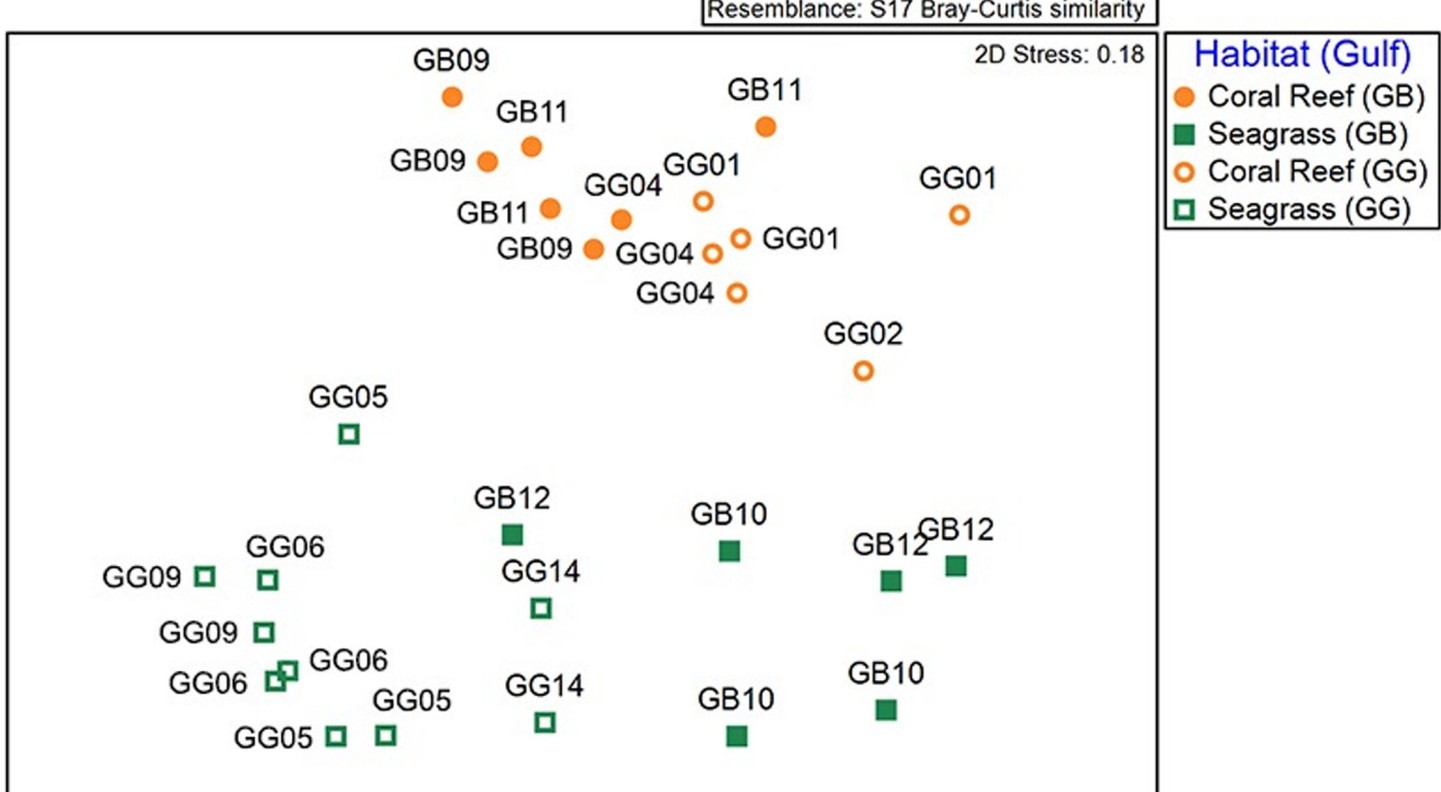

**Fig 9. Assemblage structure of mollusk death assemblages within the gulfs of Batabanó (GB) and Guanahacabibes (GG).** Similarity patterns in a NMDS ordination plot of 29 samples from 11 sites based on species abundance. Samples were coded by habitat (color and shape) and gulf (open/close symbol).

**Functional composition.** We ordered the sites along the x-axis according to their similarity based on the species abundance for better visualization of patterns. The feeding type structure showed differences related with the habitats again (Fig 10A). Sites within the reef sands (GG09, GB11, GG01, GG04, and GG02) had herbivores as the most important category, followed by carnivores. Heterogeneity in feeding type composition was larger in seagrass; some sites (GB12, GG05, GG09, and GG06) had an important contribution of chemosynthetic deposit feeders, while at GG14 dominated herbivores and at GB10 dominated suspension feeders. The relationship organism-substrate had large heterogeneity in reef sand sites (Fig 10B). Three reef sand sites (GB11, GG01, and GG04) had dominance of epifaunal forms, while at GG09 dominated infauna, and in GG02 facultative epifauna/infauna. In seagrass sites, infaunal forms were dominant, although epifauna was important at GG09. The mobility had a rather regular pattern also associated to the habitat (Fig 10C). Three reef sand sites (GB11, GG01, and GG04) had dominance of sedentary forms, while seagrass sites (GB10, GB12, GG14, GG05, GG09, and GG06) and two reef sand sites (GG02 and GB09) were largely dominated by actively mobile forms. The shell attachment had a similar pattern of habitat influence (Fig 10D). Three reef sand sites (GB11, GG01, and GG04) had dominance of bysally attached forms. The other two reef sand sites (GG02 and GB09) had important contributions of bysally attached forms, but the dominance was given by unattached forms. The latter category was largely the most important in seagrass sites.

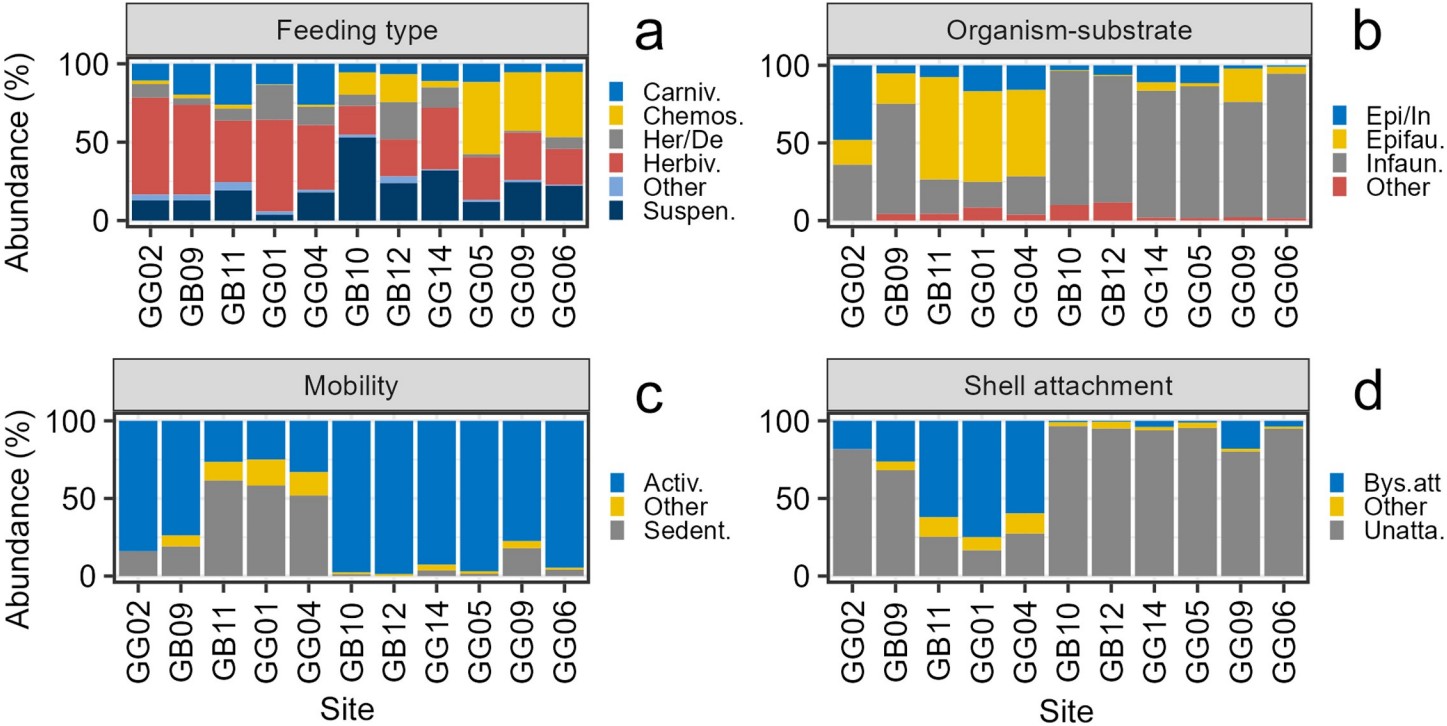

**Fig 10. Local functional composition of mollusk death assemblages in 11 sites within the gulfs of Batabanó (GB) and Guanahacabibes (GG).** (a) Feeding types including carnivore, chemosymbiotic deposit feeder, herbivore/deposit feeder (facultative), herbivore, suspension feeder, and others. (b) Relationships organism - substrate including infauna, epifauna, epifauna/infauna (facultative), and others. (c) Mobility includes actively mobile, sedentary, and others. (d) Shell attachment including unattached, bysally attached, and others. The traits (b), (c), and (d) are only for bivalves. See the text for details about "others". Note that the sites were ordered by their similarity based on species abundance.

## Discussion

Our data provide a synoptic picture of the mollusk death assemblages in two habitats within two shallow gulfs: GB in the Northwestern Caribbean Sea and GG in the Southeastern Gulf of Mexico. We detected ecological patterns related to the scale and habitat type for diversity and assemblage structure. We identified 7113 shells belonging to 393 species (94 bivalves, 290 gastropods, and nine scaphopods), that can be considered as a large sample size. However, the species accumulation curves (S1 Fig), particularly for habitats and at site scale, were hardly asymptotic suggesting that hyper-diverse assemblages may never reach an asymptote [59]. Estimates of sampling coverage were in general acceptable (i.e., > 90%) suggesting a fair description of the diversity and assemblage structure of mollusk death assemblages.

### Mollusk death assemblages at regional scale

The two gulfs showed coherent similarity in terms of species richness, β-diversity, evenness, taxonomic and functional assemblage structures (except for the feeding type); also, PERMANOVA indicated no statistical differences (S3 Table). Similar geological and biogeographical settings in the western region of Cuba Island may account for these commonalities between GB and GG. For instance, β-diversity, assemblage composition and functional structure may be explained by three general factors [60]. First, abiotic heterogeneity, which was similar between the two gulfs given the occurrence of typical tropical habitats in both basins (e.g., mangroves, muddy bottoms, seagrass meadows, and coral reefs). The abiotic data of water column and sediments, although with similar average values between gulfs, varied notably within

gulfs indicating large environmental heterogeneity (Table 1). Second, the regional pool of species should be similar in both gulfs given they belong to the same biogeographic realm (i.e., Western Tropical Atlantic, [61]). Third, the dispersal rates, which are related to species' biology and the oceanographic regime. However, it was not possible to compare both gulfs in terms of the oceanographic regime because of the lack of available data (e.g., volume of transported water, local-scale circulation).

There were two main differences between the gulfs: (i) phylogenetic and functional richness and (ii) feeding type composition. Phylogenetic and functional richness were higher in GB maybe because it has a more advantageous balance of productivity-disturbance. Richness, in the ecological time, depends on productivity and disturbance regimes [62]. The eastern portion of GB, where the four sites were located (Fig 1), is an area of high biological productivity due to the confluence of freshwater (and nutrients) runoff from Zapata Swamp and the influence of the Caribbean current on the shelf [63]. Local upwellings, eddies and tidal currents have been reported to promote the exchange between the open ocean and the southeastern portion of GB [64]. Additional evidence of the large productivity of eastern GB is the healthy coral reefs status with coral colonies of large size [31] and the high finfish and spiny lobster catches [65]. Besides, the disturbance regime seems to be high in the shelf border of eastern GB. High contribution of the bysally attached bivalve *Barbatia domingensis* adapted to strong currents and the occurrence of skeletal remnants of oceanic holoplanktonic species suggest stronger influence of open ocean. In addition, reef sands in GB exhibited the largest number of shells (and total species richness) suggesting high secondary productivity (S2 Fig).

The feeding type composition was different between GB and GG, with higher importance of chemosynthetic bivalves (Lucinids) in the latter. These bivalves are dominant in sulfide-rich hypoxic sediments [66], like the four seagrass sites located in the inner part of GG. On the contrary, the seagrass sites in GB did not show symptoms of hypoxia (M. A. personal observation) and likely oxygen replenishment occurred due to intense hydrodynamics.

Two additional regional patterns were related to β-diversity. The first pattern was derived from the comparison of β-diversity dimensions (i.e., species versus phylogenetic versus functional). The species β-diversity was the highest (about 0.6, Fig 2B) because of the high number of rare species promoting differences between sites. Functional β-diversity was the lowest (about 0.3, Fig 2F) likely indicating that taxonomically different species were replaced by functionally similar trait combinations. While we are not aware of reports about functional β-diversity for mollusk assemblages, similar findings were reported for diatom assemblages [67]. Phylogenetic β-diversity was intermediate (about 0.5, Fig 2D) because composition at higher taxonomic level (i.e., genus, family, etc.) indicated a balance between species rareness and similar functional trait combinations.

A second regional pattern of β-diversity, for the three diversity dimensions, was the larger contribution of the replacement partition compared to the richness partition. Legendre [18] stated that replacement is related to ecological tolerance or niche breadth, while richness depends on the number of niches. Our findings broadly confirmed this statement because many species with potentially different niche breadth occurred in both gulfs supporting a high replacement partition of β-diversity. Meanwhile, our sampling scheme only included sediment-related niches that could explain the lower contribution of the richness partition to β-diversity.

We recorded high evenness of the assemblages at regional scale, with the presence of one moderately abundant species and a very long tail of rare species (Fig 3A). This pattern agrees with the very high richness and low dominance of mollusk coastal assemblages in Cuban Archipelago as reported by other authors [33–35,37]. The high evenness reported in our study is also enhanced by two features [5]: (1) time averaging increases the probability of several

shifts in species dominance through time resulting in higher average evenness, although we have evidence of limited time averaging in our study. And (2), at regional scale there is more tendency to reflect the shared dominance of multiple species within a window of time, resulting in lower dominance.

## Mollusk death assemblages: Reef sands versus seagrass meadows

The habitat type had a strong influence on the diversity and assemblage structure, which agrees with reports about dead [10,68,69] and live mollusks [28]. Reef sands harbored a high richness of mollusks, which also agrees with other studies (e.g., [70,71]). The balance between productivity and disturbance [62] may explain our findings. Coral reefs function upon high quality organic carbon, which is eventually available to consumers (e.g., herbivore or carnivore mollusks) [29]. Also, in coral reefs the physical disturbance by waves and currents reduces oxygen depletion, which is a limiting factor for mollusks. Seagrass meadows harbor a large biodiversity as well, but their balance productivity-disturbance is likely less favorable compared to reef sands explaining the lower richness. For instance, an important fraction of organic carbon in seagrass is highly refractory due to the content of lignin resulting in less food availability for herbivore mollusks [72]. Lower mollusk richness in seagrass meadows could also be related to less physical disturbance that reduces oxygen replenishment causing hypoxia and the generation of hydrogen sulfide [66].

Each habitat had a distinctive assemblage structure suggesting that the information about abundance and diversity can be preserved long after death; this agrees with Casebolt and Kowalewski [4]. This finding is evidenced by the differences in richness and adaptative traits, but also suggested a rather limited shell transport between adjacent habitats. The transport of dead shells is an important postmortem process [5,73], but likely the physical structure of the habitats diminishes its magnitude. Seagrass meadows are physically protected environments because the canopy and rhizomes may reduce the erosion of sediments by currents and waves [72]. Meanwhile spur-and-grooved structures in coral reefs are also physically sheltered by rocky reef structures [74]. Albano and Sabelli [75] reported similar findings about depleted lateral shell transport in seagrass meadows and coral reefs in a study about the fidelity of live versus death assemblages.

The functional structure was different between the two habitat types probably because of niche preferences. Reef sands were dominated by epifaunal herbivores adapted to graze on the microphytobenthos; in particular, herbivore gastropods contributed importantly to this guild. In sandy spots, microphytobenthos has a large primary productivity and provides substantial food availability to the benthic consumers [76,77]. Interestingly, suspension feeder bivalves were not particularly abundant in reef sands, likely indicating low availability of suspended particles. Sedentary species, mostly bysally attached bivalves, also thrived in reef sands suggesting an adaptation against the resuspension by currents. In seagrass meadows, chemosynthetic detritivores were more important, pointing that much of the primary production done in the seagrass canopy reaches the bottom as detritus. Seagrass meadows were different between GB and GG due to the higher contribution of suspension feeder and chemosynthetic bivalves, respectively. Suspension feeders dominated GB's seagrasses probably because of the intense hydrodynamic regime in the shelf border. Current velocity exerts a physical disturbance on herbivores, although it is counteracted by the seagrass´ canopy [78]. In GB's seagrasses, the high shoot density reported by us (Table 1) possibly enhanced the protection against resuspension promoting the dominance of unattached forms. Meanwhile chemosynthetic lucinid bivalves were more abundant in GG because slower hydrodynamic reduces oxygen replenishment and promotes hydrogen sulfide generation in sediments [66]. Infaunal/actively mobile/

unattached forms were abundant in all seagrass meadows likely because the individuals´ mobility is an advantage in the search for oxygenated spots and/or less refractory detritus.

## Mollusk death assemblages at local scale

Local species richness indicated the combined influence of the regional pool of species [79] and habitat type [28]. In our study, between-site richness variability within the same gulf and habitat type depended on the local environment. For instance, the highest richness and abundance at GB11 and GG04 responded to local environmental conditions that likely enhanced the deposition of shells in sandy spots within spur-and-groove structures.

The total β-diversity at local scale showed moderate shifts from site to site likely because of differences in abiotic heterogeneity [60]. In terms of β-diversity partitions, many different species with potentially different niche breadth may explain the higher replacement partition across sites [18]. Conversely, a similar number of niches across sites within the same habitat explains the lower contribution of richness difference partition.

Evenness curves indicated that sites GG06 and GG09 were subjected to disturbance because of the dominance of a few hypoxia-tolerant species of *Parvilucina*. This agrees with Armenteros et al. [37] that reported significant decline of mollusk diversity at the same sites based on core stratigraphy and $^{210}$Pb dating. The pattern of similarity among replicates in the NMDS (Fig 9) suggested the overarching effect of habitat type on the assemblage structure, which agrees with other studies (e.g., [70]). Nevertheless, substantial variability among replicates within a site occurred probably as response of the assemblages to small-scale sediment heterogeneity [80–82]. Indeed, in our study the factor SITE only explained about one third of the total variance (S3 Table).

## Effects of time averaging on the diversity outcomes

The time averaging partially explained the large richness of the mollusk death assemblages in our study [5]. Instead of a bias, the time averaging can be considered as a scaling post-mortem process that averages the input of shells over time [2]. Thus, we can expect that our death assemblage study may inform more about biodiversity than a single snap-shot study using only the living mollusks. We revised the $^{210}$Pb age models published in Armenteros et al. [37] from the same seagrass sites in Guanahacabibes looking for evidence about time averaging in our study. These age models were based solely on sediments, but they have the advantage that samples were taken at nearby sites (< 100 m) using a barrel corer that effectively samples the top 10 cm of sediments [37]. A limitation is that estimations of the stratigraphic resolution and time averaging in the Holocene record ultimately requires the coupling of age models and post-mortem age-frequency distributions of dated shells [3]. Nevertheless, simple age models derived from sediment cores still can be informative about the temporal scale of fossil assemblages even in the absence of post-mortem age-frequency data [3].

We estimated that the temporal framework in our study was about 100 years because the top 10 cm of sediments at sites GG05, GG06, and GG09a spanned a period of 58, 99, and 81 years, respectively [37]. This supports the value of death assemblages to analyze the diversity of modern mollusk assemblages. We identified other indirect lines of evidence supporting this rather short time averaging of about one century. First, the sedimentation rate in GG was high [37] promoting as consequence a low time averaging, regardless the mixing depth and rate of disintegration [3]. Second, the large taxonomic-, phylogenetic- and functional β-diversity suggested a limited homogenization of assemblage composition due to time averaging [2,68]. Lastly, our evidence also indicated that time averaging was lower in seagrass meadows compared to reef sands due to lower reworking and higher sedimentation rate, this agrees with

Kidwell [1]. In reef sands the time averaging probably spanned a longer period because of the stronger sediment mixing and lower sedimentation rate given by intense hydrodynamics [5].

## Conclusions

Our study highlights the high diversity of mollusk death assemblages in inner seas from Northwestern Caribbean Sea and Southeastern Gulf of Mexico. The integration of taxonomic, phylogenetic, and functional information is a powerful approach, but still unexplored for some hyper-diverse marine taxa such as mollusks. We discovered spatial patterns at regional and local scales, although habitat type had the stronger effect on the diversity and assemblage structure. Reef sands had higher richness than seagrass meadows supporting that coral reefs are hotspots of marine biodiversity. Phylogenetic and functional composition showed evolutionary adaptations to the habitat type. Specifically, reef sands were characterized by epifaunal herbivores likely adapted to feed on microphytobenthos and bysally attached bivalves adapted to intense hydrodynamics. In seagrass meadows, the hydrodynamic regime influenced the functional structure, with dominance of suspension feeders in exposed sites and dominance of chemosynthetic infaunal bivalves where oxygen replenishment was limited. Our study highlights the convenience of including phylogenetic and functional traits, as well as dead shells, for a more complete assessment of mollusk biodiversity.

## Supporting information

**S1 Fig. Species richness and β-diversity accumulation curves.** (a) Regional species richness. (b) Regional β-diversity. (c) Habitat species richness. (d) Habitat β-diversity. (e) Local species richness Dots indicate the observed richness, shaded areas the 0.95 CIs.
(TIF)

**S2 Fig. Regional composition.** Heat map of relative abundance (in %) of those typical species that contribute to the 70% of similarity within each gulf. Note that species were ordered by their similarity across the gulfs.
(TIF)

**S1 Table. Matrix of biological traits of mollusk species.**
(XLSX)

**S2 Table. Matrix of mollusk species per sample.** The replicates are denoted as R1–3.
(XLSX)

**S3 Table. Results of the Permutational Analysis of Variance (PERMANOVA).**
(DOCX)

## Acknowledgments

R Peraza-Escarrá thanks to PhD Graduate Program "Ciencias del Mar y Limnología" at Instituto de Ciencias del Mar y Limnología (UNAM) and to Consejo Nacional de Humanidades, Ciencias y Tecnologías (CONAHCYT). We acknowledge the crew of the R/V Felipe Poey. We thank the Centro de Estudios Ambientales de Cienfuegos, particularly Misael Díaz-Asencio and Lisbet Díaz-Asencio, for sediment analyses. We thank Dania Saladrigas and Lorena González for sample processing, Abel Valdivia for providing the map, and Fernando Bretos and Daria Siciliano for their support in the field and technical assistance. The collection of sediment samples in Cuban waters was made under the permit LH112-AN-(23)-2014 granted by Centro de Inspección y Control Ambiental (CICA). We acknowledge very much the

comments by Susan Kidwell, Danae Thivaiou and one anonymous reviewer that improved the manuscript.

## Author Contributions

**Conceptualization:** Maickel Armenteros, Adolfo Gracia.

**Data curation:** Rosely Peraza-Escarrá, Raúl Fernández-Garcés.

**Formal analysis:** Rosely Peraza-Escarrá, Maickel Armenteros.

**Funding acquisition:** Maickel Armenteros, Adolfo Gracia.

**Investigation:** Rosely Peraza-Escarrá.

**Resources:** Maickel Armenteros, Adolfo Gracia.

**Supervision:** Maickel Armenteros, Adolfo Gracia.

**Writing – original draft:** Rosely Peraza-Escarrá, Maickel Armenteros, Adolfo Gracia.

**Writing – review & editing:** Rosely Peraza-Escarrá, Maickel Armenteros, Raúl Fernández-Garcés, Adolfo Gracia.

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
