## [Decision Letter · Decision Letter 0]

27 Nov 2023

PONE-D-23-26159Diversity patterns of mollusk death assemblages in coral reefs and seagrass meadows and a comparison with living assemblagesPLOS ONE

Dear Dr. Peraza-Escarrá,

Thank you for submitting your manuscript to PLOS ONE. After careful consideration, we feel that it has merit but does not fully meet PLOS ONE’s publication criteria as it currently stands. Therefore, we invite you to submit a revised version of the manuscript that addresses the points raised during the review process.

We look forward to receiving your revised manuscript.

Kind regards,

José A. Fernández Robledo, Ph.D.

Academic Editor

PLOS ONE

Journal Requirements:

Did you know that depositing data in a repository is associated with up to a 25% citation advantage (https://doi.org/10.1371/journal.pone.0230416)? If you’ve not already done so, consider depositing your raw data in a repository to ensure your work is read, appreciated and cited by the largest possible audience. You’ll also earn an Accessible Data icon on your published paper if you deposit your data in any participating repository (https://plos.org/open-science/open-data/#accessible-data).

"R Peraza-Escarrá thanks to PhD Graduate Program “Ciencias del Mar y Limnología” at Instituto de Ciencias del Mar y Limnología (UNAM) and to Consejo Nacional de Humanidades, Ciencias y Tecnologías (CONAHCYT) for a PhD scholarship (CVU_1080567). M Armenteros thanks to CONAHCYT for a postdoctoral fellowship (CVU_982475). This research was supported partially by The Ocean Foundation through the “Proyecto Tres Golfos”. We acknowledge the crew of the R/V Felipe Poey. We acknowledge the Centro de Estudios Ambientales de Cienfuegos, particularly Misael Díaz-Asencio and Lisbet Díaz-Asencio, for sediment analyses. We thank Dania Saladrigas and Lorena González for sample processing, Adrián Martínez for providing the map, and Fernando Bretos and Daria Siciliano for their support in the field and technical assistance. The collection of sediment samples in Cuban waters was made under the permit LH112-AN-(23)-2014 granted by Centro de Inspección y Control Ambiental (CICA)."

Funding information should not appear in the Acknowledgments section or other areas of your manuscript. We will only publish funding information present in the Funding Statement section of the online submission form. 

"RPE received support from a PhD scholarship (CVU_1080567) granted by the Consejo Nacional de Humanidades, Ciencias y Tecnologías (CONAHCYT). MA received support from a postdoctoral fellowship (CVU_982475) also granted by CONAHCYT. Additional funds were obtained by The Ocean Foundation through the “Proyecto Tres Golfos”. The funders had no role in study design, data collection and analysis, decision to publish, or preparation of the manuscript."

5. We note that Figure 1 in your submission contain map images which may be copyrighted. All PLOS content is published under the Creative Commons Attribution License (CC BY 4.0), which means that the manuscript, images, and Supporting Information files will be freely available online, and any third party is permitted to access, download, copy, distribute, and use these materials in any way, even commercially, with proper attribution. For these reasons, we cannot publish previously copyrighted maps or satellite images created using proprietary data, such as Google software (Google Maps, Street View, and Earth). For more information, see our copyright guidelines: http://journals.plos.org/plosone/s/licenses-and-copyright.

(1) You may seek permission from the original copyright holder of Figure 1 to publish the content specifically under the CC BY 4.0 license.  

**Additional Editor Comments:**

Dear Dr. Peraza-Escarrá

Both reviewers have done an extensive analysis of the manuscript with numerous suggestions. Although they disagree on the final evaluation, please address the comments expressed by both of them.

Reviewers' comments:

Reviewer's Responses to Questions

**Comments to the Author**

1. Is the manuscript technically sound, and do the data support the conclusions?

Reviewer #1: Partly

Reviewer #2: Partly

2. Has the statistical analysis been performed appropriately and rigorously? 

Reviewer #1: Yes

Reviewer #2: N/A

3. Have the authors made all data underlying the findings in their manuscript fully available?

Reviewer #1: Yes

Reviewer #2: Yes

4. Is the manuscript presented in an intelligible fashion and written in standard English?

Reviewer #1: Yes

Reviewer #2: Yes

5. Review Comments to the Author

Reviewer #1: This ms is well-organized and well-written ms, and addresses some important, mostly still-understudied issues in the application of molluscan dead-shell assemblages to the analysis of marine biodiversity. It includes:

(1) a spatially-nested analysis of the diversity (several measures) and structure (not fully clear = relative abundance distribution?) of death assemblages; really nice design, and so great to see new field data from this important part of the Caribbean, on top of the general issues;

(2) an analysis of molluscan death assemblages as a surrogate of the combined dead+living molluscan assemblage; analogous to how benthic foram people approach field samples owing to the challenges of differentiating living and dead individuals, not clear why the authors would do this rather than a more conventional dead versus live comparison, or at least to the exclusion of a more conventional analysis (were sample sizes of living mollusks too small?); and

(3) a test and quantification of the extent of down-slope transport of shelf-dwelling species into deep-sea, i.e. slope, sediments; wonderful to have, for comparison with the few other strong tests of this, most notably field data from Bouchet in New Caledonia, and then a zillion anecdotal-quality observations (summarized 30y ago in Kidwell/Bosence).

The paper thus covers a *lot of intellectual territory and has a rather large number of figures (some could probably go into an appendix, and simply be compared using text to a smaller number of figs published in the main text). But it is well-referenced throughout, connecting the findings well with the large existing literature, both observational and theoretical.

As above, my concerns and suggestions are interspersed below among features that I really like, and were written as I initially read the ms. Overall, I think this would make an excellent paper in Plos-1 and recommend acceptance with only modest revisions, should the editor agree.

Reviewed by Susan Kidwell, thanks for the opportunity.

Methods = field collection methods were actually better than the ‘usual’, with authors using box cores rather than van Veens, which are the standard, although, in the end, the depth of penetration of this gear was still only ∼10 cm. They thus likely missed collecting many larger-bodied, adult, and especially deeper-burrowing taxa – this is a common problem, should be mentioned either here in Methods or in the Dsicussion, as a ‘caveat’, but the authors need not be defensive about it – although it biases the sample, it is an almost universal bias, and thus permits comparison to the many many other, similarly-limited datasets (see Lees 2021 Neth J Sea Res, and Powell &Mann 2016 Cont Shelf Res for discussions). I especially admire that they acquired these samples on SCUBA, allowing for careful ad known placement… and 3 replicates per site. Description of field study areas is exemplary – these authors really describe the areas in useful ways, including extent of physical disturbance and, of course, likely human effects, drawing on earlier work (includes the admirable Table 1)… thus well-chose field areas. The 0.5mm sieve is much finer than is usual (1mm, and in some live-dead studies by geologists/paleoecologists, a 2mm or coarser sieve is used) – only a concern because this likely included juvenile individuals that can be hard to identify AND have unlikely persistence, both as living individuals (persistence to adulthood) and as dead shells (postmortem) – although I appreciate that tropical faunas can have many many ‘micromollusks’ that might be largely missed with a 1 mm mesh. Just something worth mentioning, as a guide to other researchers thinking of repeating this kind of analysis elsewhere.

Live data have already been reported, in Armenteros et al 2018 – those results should be provided in tabular form here because the sample sizes of this living assemblage is critical to evaluating live-dead agreement

Analysis – Richness: I really like the authors’ approach for taxonomic richness, strongly prefer accumulation curves rather than post-hoc rarefaction fittings, AND I like using effective numbers of species for richness – super. Phylo richness – not familiar, feels like a stretch to add this, but no objection. Functional richness – good use of Todd & Mikkelson/Bieler for coding taxa; although the N of categories is (slightly) smaller than the S of species, could still use the same metrics; but I like bringing in these FD metrics from biogeography/macroecology. Beta – many ways to do it, they should also consider (and in discussion compare their results to) the methods used in Tomasovych/Kidwell 2009-2011 (e.g., diversity transfer from alpha to beta, spatial gradient capture, richness/evenness at alpha) – a series of papers by those authors (summarized in Kidwell Tomasovych 2013) explore the effects of time-averaging on the suite of metrics used here, seems to be one part of the paleo/taph literature missed. Evenness – would prefer to see a quantitative metric of evenness, such as the sample-size-independent Hurlbert’s PIE, rather than only a graphical depiction; N singletons and doubletons is a measure of the proportion of rare taxa, which is useful/important, but not the same as dominance/evenness that focus on the proportional abundances of the top few taxa and overall shape of the cumulative richness curve. PIE is now widely used in live-dead analysis.

Ah—structure = simper analysis, not what I’d call structure, but rather inter-sample similarities, series of pairwise comparisons…

Comparison of live-dead – using the ancient, very simple metrics of Kidwell & Bosence 1991. That’s OK as long as include a table with the raw N’s being compared (e.g., how similar is the N of living and dead individuals?), because those ancient authors found that the small-sample size of the living assemblges in most studies was the most important factor explaining (correlated with) the low apparent live-dead agreements (which the later modeling of Kidwell and others showed was true). But a full live-dead nowadays would explore live-dead using standard ecological measures such as (dis)similarity of paired live-dead species lists for a site/habitat/region, ditto in rank-abundance of taxa live&dead, nmds etc. [I do note a NMDS lines469; another use of the word ‘structure’, here meant to indicate *spatial variation in species composition of assemblages, whereas structure also used to indicate RADs… be careful, would be good to sharpen language

**I fear that the omission of such measures owes to very low abundances of living individuals, an especially large challenge in tropical settings in my experience… typically we have to process far more sediment for living than for dead to achieve statistical power (see Powell et al from the early 1980s for the first time this was recognized and acted upon; Gilad et al 2018 did the same thing, sieving many-x more volume of sediment for live than dead, in tropical shelf habitats for the same reason). If this is indeed the situation, the authors, again, do not need to be defensive but simply straightforward and ‘up-front’, that is explain from the start that there are limits to what they can do analytically given extremely disparate live-dead sample sizes (and probably extremely small Ns of living in absolute terms, e.g. anything <50 inds and especially <20 inds).

Results – first para on general features is where the authors should immediately start differentiating their living and death assemblages – e.g., of these 7113 inds, how many dead & how many alive, ditto the S of species orders etc. If all of these numbers and others in the Results section are going to refer excluseively to the death assemblages, then this also needs to be made explicitly clear here, e.g. “All results refer only to dead-shell assemblages unless otherwise noted (e.g., section xx on dead vs total comparison).” I find it hard to proceed without knowing this basic information

Line 333 – ‘fluctuations’ typically would refer to variability in time, whereas here I think the authors mean variability among spatial replicates at a site etc collected during the same season… watch for this

Line 514 Table 2 – Finally! Info on N live and N dead… and indeed the disparity is super high, not shocking for a tropical shallow marine system but very troubling for live-dead comparison. My suspicion re why the authors chose to compare dead versus live+dead are confirmed. This is not a scientifically fair/appropriate test of the ability of the dead to serve as a surrogate of the usual, entirely-live-collected kind of data that biologists typically have – and moreover flips the usual disparity (usually the ‘whole’ is much larger/broader/more inclusive sampling of the target than is the subset being tests as a ‘surrogate’/proxy of i). The results are of course interesting, but mostly for enlightening readers about the challenges of live-dead analysis in such settings. I think the discussion and conclusions from this section will thus need to be very modest, very modulated. ** For the editor: The disparities are e.g. 1500 dead inds versus 5 living ones, etc, with only two sites yielding >20 individuals, with 20 being an absolute minimum N of individuals to do any kind of nmds etc.

In this table 2, I’m most intrigued by the (to me) large number of taxa found exclusively alive, i.e. “alive-only” – I would not expect this given the huge disparity in sample size, i.e. that any taxa would not be encountered among the dead. 2 explanations = (1) new, invasive taxa, populations still small, little opportunity for dead shells to accrue to measurable levels in the death assemblage [given the species, I doubt this, but authors should (in the discussion) indicate if these are all natives or otherwise commonly encountered already by others in Cuba; or (2) taphonomic bias, i.e. these are all taxa with truly exceptionally low preservation potential, e.g. from their especially small/thin-shelled bodies, and/or super-rarity, and/or high-organic aragonitic shell types, and/or life-habit (e.g. I find that commensals have very low preservation potential). From my experience, Solemya has a super-low pres pot (as do lyonsiids, pinnids, some pandoriids), and I commonly find that Bittium, Caecum, and Rissoids/small parasitic & minute herbivores are strongly underrepresented in the dead (but not actually absent) Limaria and Americardia & Codakia should definitely not fall into this category… anyway, a surprising and super-interesting result

523 contribution of shelf species to deep-sea death assemblages – wonderful, super useful observations. It would be super-helpful to provide the typical or maximum known body sizes of these taxa found off shelf, and/or, even better, the typical or max body sizes observed among thte individuals actually collected. Would make a more powerful discussion/conclusions

Table 3 this section – Please provide info at the bottom of each column on the total N of the death assemblage so that readers can get full value from these prop abundance values. E.g. does 100% (“1”) mean 10 out of 10, 100 out of 100 inds, or 1 out of 1? The data on raw N of dead shells from a tropical slope would be valuable in itself.

=Interesting from a taphonomic perspective that all are gastropod species w one exception… because epifaunal, which is long thought to increase the potential for postmortem transport (because they die above the sediment-water interface—idea probably in KidwellBosence)? Some info on life habit, not just gastro-bivalve class membership, would be nice to add to this table, or at least to bring into discussion.

Discussion –

549 hyper diverse, may never reach an asymptote – probably true, although still not a popular idea, but that’s because most people have never encountered the tropics in a first-hand way. Thus am glad to see you say this, is a perfectly good description of your curves!

Regional scale – agree w your interpretation

579 – again, people generally don’t like to talk about high productivity and coral reefs, or the role of upwelling, but am sure this is a factor in a lot of places and here you have independent evidence of upwelling to promote it; nice regional insight

614 – yes Kidwell/Bosence for the even-ing effects of time-averaging (also see the older Peterson paper they cite), but there’s more explicit, rigorous testing of this effect in younger literature, e.g. Olszewski/Kidwell 2007, var Tomasovych/Kidwell (it’s also in SK/ATomas 2013).

620 habitat – by the way, when I first saw ‘coral reef’ in the abstract, I assumed this would be an analysis of mollusks in the reef framework itself (with all of those huge challenges, e.g. see many nice papers by Zuschin). So I was happy/relieved to see that the authors actually meant the sandy seabeds between reef tongues, i.e. the grooves… be sure that this is clear in the abstract and intro, and then again remind reader in the discussion. Lots of potential for shells washed in from adjacent hard substrata/reef framework, but also plenty of taxa that live (and die) in the sands themselves.

640ll – yes, transport *can be important, but commonly less that folks worry about it being; thanks for having read the SK/Bosence paper carefully enough to realize that we concluded it was a relatively minor issue in most level-bottom subtidal settings (ditto even on steep shelves per SK 2008, right!).

657 – indeed, it *is interesting that susp feeders are as unimportant as they are in these sandy inter-reef areas �oh, that’s a term you could use, “inter-reef sand” rather than “coral reef”

To 672 – I really love all the detailed info and thinking about the natural history of your settings, it’s a usefully thoughtful discussion

687 – evenness curves – OK, not crazy about this approach

693 – ‘substantial inter-replicate variability, i.e. small-scale spatial heterogeneity – I find this, too, in tropical shelves… the world is patchier than we imagine, even with time-averaging. Am glad that the authors are being ‘true’, i.e. fully honest about their findings, are not trying to smooth it over

Live-dead assemblages

Tomaso et al 2023 – no need to be defensive, this is an ‘ideal’ work plan, very difficult and expensive to achieve! BTW Adam and I try very hard to get other workers to appreciate ‘time averaging’ as a scaling process, which operates post-mortem, but NOT as a taphonomic process the way people typically mean that. T-avg does not ‘bias’ the record, but rather simply averages the input of shells over time, with a series of predictable effects (as summarized SK/AT 2013). It provides a window of opportunity for taph processes such as transport, fragmentation, bioerosion, etc to genuinely bias the composition of that time-averaged assemblage. But time averaging simply means we’re sampling bio over a period of time longer than the typical snap-shot of a biological survey – if one were to sum many live-collections over time, the effects onn richness, evenness would be the same, *unless one biased that assemblage by e.g. removing many of the fragile-shelled species. Sorry to be this pushy, but the authors are careful thinkers and ambitious, so it would be great to have the time-averaging vs taphonomic-bias language sharpened in this ms.

710—ah, Table 2 discussion! Ok, I see your very good discussion, thanks.

737 – delighted to see this thoughtful discussion of time averaging, AND that the authors proceed to think-through their local situation given the 210Pb data at hand. Logical, fine job.

757 – and dead-only treatment, good. I think they’re completely right… although Kidwell/Bosence is really old, we hit upon a fundamental truth

Shelf-deep-sea spp –

I wondered about changing “deep-sea” to “slope”, throughout the ms, but 1500m is a really deep part of a slope, so just leave it alone. Nice discussion

Congratulations on a really nice piece of work – a *lot of fieldwork, a *lot of lab work to pick/sort/count these dead individuals down to such a small body size… and then looking at so many metrics including some still-novel ones for live-dead analysis, e.g. FD and Phylo-D.

With best wishes, SK

Reviewer #2: The manuscript " Diversity patterns of mollusk death assemblages in coral reefs and seagrass meadows and a comparison with living assemblages." authored by Peraza-Escarrá and collaborators evaluated the diversity patterns of mollusks assemblages in two gulfs and two different habitats. The main questions of the study are to (1) investigate the taxonomic, phylogenetic, and functional diversities of mollusk death assemblages at regional and local scales in 19 coral reefs and seagrass meadows; (2) evaluate death assemblages as surrogates of the whole mollusk assemblages (alive + dead); and (3) explore the downslope transport of shelf species to the deep sea. The manuscript is relatively well-written but excessively lengthy. The topic is very interesting to the scientific community, but some issues should be addressed, and missing information should be included before publication.

My main concerns are about how the manuscript is organized, it gives the impression of being organized more like a chapter of a Ph.D. thesis than a scientific article, including too many details often irrelevant to the manuscript's fundamental topic. The main goal is confusing since the title highlights the “mollusk death assemblages in coral reefs and seagrass meadows and a comparison with living assemblages” and there is no specific development of this idea/topic. For example, different analyses with alive and death assemblages separately could be interesting for differentiating them and could be interesting in terms of current ecological differences between seagrass meadows and coral reef mollusks. Finally, for the proposed experimental design is also necessary to provide more specific details that are not yet included.

The paper in its current form needs major revision before publication. Below I provide some suggestions to improve the manuscript.

Specific comments

The authors should carefully review and choose the article's topic and then select the important and relevant information on that topic (seagrass meadows vs coral reef?), discarding the rest. In its current form, I consider the manuscript to be too lengthy, containing a wealth of information without a clear guiding thread. On the other hand, it appears to be a mix of data and interpretations in which, at times, it is difficult to find a common thread. Once the authors focus on the objective of what they want to convey in the article, they should direct the discussion to that topic. It's worth noting that the bulk of the work has already been done, as this should only involve a readjustment and reorganization of content.

A thorough review of the English is necessary to correct mistakes.

Abstract

Authors should integrate better the sentences, sometimes seems to be a succession of sentences without links between them

Introduction

The authors should rearrange the introduction. I consider it very important to include information about coral reefs and seagrass meadows in the ocean and specifically the mollusk assemblages in those habitats, and why is very important to study them.

Authors should reorganize the introduction from the most general to the most specific.

Material and Methods

Why isn't there the same number of replicates in all cases? Because in the case of GB, we have two seagrass sites and two coral reef sites, but in the case of GG, there are 3 and 3? Why wasn't the same number of replicates (n=3) maintained in all cases? Justify it

Give more details of the deep sea sampling

Is a single slope replicate sufficient to address the differences between the communities of the platform and the slope? Shouldn't the study be balanced? Justify it because this fact could compromise the entire comparison

Authors should also consider to conduct multivariate analyses such as PERMANOVA that assess the entire community and separately for living and dead mollusk assemblages.

It should be considered by the authors to give more details of the experimental design used, specifying the fixed factors, random factors, nested and orthogonal factors.

Results

At times, it is stated that there are significant differences between certain factors. What statistical tests have been conducted to assert the existence of such differences?

The title refers to the comparison of living mollusks with the dead ones, but this comparison is not explicitly addressed except in a small section discussing percentages. Furthermore, throughout the manuscript, there is no mention of a separate matrix that differentiates these two distinct communities

Some parts of the results are redundant and some are unnecessary (depending on the topic of the manuscript); for instance, the information provided in Figures 4 and 7 is essentially the same. It would make more biological sense to treat it separately by habitats as it appears in Fig 7 (seagrass and coral reefs) according to the indicated in the title of the work, rather than jointly in GB and GG. The authors should review this idea and provide only the necessary results for the study's topic.

Perhaps it's a good idea to analyze the data in a more general manner without delving into details of differences between replicates and specific gulfs. If the goal is to contribute to the broader scientific community, it might be more important to focus on observable differences in mollusk assemblages between habitats (seagrass vs. coral reef). This approach could involve explaining differences in species number, abundance, and functional traits rather than detailing local differences between the two. With this shift, the almost exclusive local focus of the study can be broadened to a much more global interest

Discussion

Considering the aforementioned points, it might be advantageous to reframe the discussion. Instead of delving into intricate details of differences between replicas and specific gulf regions, a more effective approach could be to analyze the data in a broader context. Focusing on the overarching differences in mollusk assemblages between habitats, specifically comparing seagrass and coral reef environments, would align better with a global scientific community. This approach would involve elucidating variations in species number, abundance, and functional traits, steering away from intricate local disparities. By adopting this perspective, the study's interest could transition from predominantly local to a more globally significant contribution

Figures and Tables

The authors should integrate the Figures and Tables in the manuscript as closely as possible to the reference in the text to facilitate the reading of the manuscript.

The authors should keep only relevant figures and tables, taking in mind that this is a scientific paper rather than a chapter where considering and including all the possible information is necessary and mandatory

6. PLOS authors have the option to publish the peer review history of their article (what does this mean?). If published, this will include your full peer review and any attached files.

Reviewer #1: **Yes: **Susan Kidwell

Reviewer #2: No

---

## [Author Response · Author response to Decision Letter 0]

31 Jan 2024

Dear editor and reviewers:

Thank you very much for the sound evaluation of our paper. We appreciate your suggestions and criticisms very much, we carefully read all of them and accepted the majority. In agreement with reviewer # 2, the paper has been shortened by one third approximately. Also based on suggestions from both reviewers, we removed specific objectives 2 and 3 because of the small number of living mollusks and single deep-sea site, respectively.

Below, you can find a detailed response to each point. For easier reading, we have split paragraphs with a hyphen when specific responses are required, and our responses start with R/.

Looking forward for a positive response for this new version,

Rosely Peraza (on behalf of other authors)

Review Comments to the Author

Reviewer #1:

- This ms is well-organized and well-written ms, and addresses some important, mostly still-understudied issues in the application of molluscan dead-shell assemblages to the analysis of marine biodiversity. It includes:

R/ Thank you very much for your opinion, it means a lot to us.

- (1) a spatially-nested analysis of the diversity (several measures) and structure (not fully clear = relative abundance distribution?) of death assemblages; really nice design, and so great to see new field data from this important part of the Caribbean, on top of the general issues;

R/ Thank you. We revised all instances of the use of “community structure” in the manuscript to avoid confusion, if possible, we used a more specific term (e.g., composition).

- (2) an analysis of molluscan death assemblages as a surrogate of the combined dead+living molluscan assemblage; analogous to how benthic foram people approach field samples owing to the challenges of differentiating living and dead individuals, not clear why the authors would do this rather than a more conventional dead versus live comparison, or at least to the exclusion of a more conventional analysis (were sample sizes of living mollusks too small?); and

R/ Indeed, that was the case, living mollusks were too few, then the comparison dead vs live was highly biased because the disproportionate sample size of the two data sets. Based on your comments and the ones by reviewer # 2 about the length of the manuscript, we decided to remove the analysis of dead versus live mollusks (specific objective 2).

- (3) a test and quantification of the extent of down-slope transport of shelf-dwelling species into deep-sea, i.e. slope, sediments; wonderful to have, for comparison with the few other strong tests of this, most notably field data from Bouchet in New Caledonia, and then a zillion anecdotal-quality observations (summarized 30y ago in Kidwell/Bosence).

R/ We agree with reviewer # 2 about that the extent of down-slope transport based on a single deep-sea site is hard to evaluate beyond doubt. Therefore, we removed the specific objective 3. 

- The paper thus covers a *lot of intellectual territory and has a rather large number of figures (some could probably go into an appendix, and simply be compared using text to a smaller number of figs published in the main text). But it is well-referenced throughout, connecting the findings well with the large existing literature, both observational and theoretical.

R/ We appreciate and acknowledge your respected opinion. Following your advice, we moved those figures do not convey core information to supplementary material.

- As above, my concerns and suggestions are interspersed below among features that I really like, and were written as I initially read the ms. Overall, I think this would make an excellent paper in Plos-1 and recommend acceptance with only modest revisions, should the editor agree.

Reviewed by Susan Kidwell, thanks for the opportunity.

R/ Thanks again for your encouraging words, your work has inspired us in our studies about mollusk death assemblages.

- Methods = field collection methods were actually better than the ‘usual’, with authors using box cores rather than van Veens, which are the standard, although, in the end, the depth of penetration of this gear was still only ∼10 cm. They thus likely missed collecting many larger-bodied, adult, and especially deeper-burrowing taxa – this is a common problem, should be mentioned either here in Methods or in the Discussion, as a ‘caveat’, but the authors need not be defensive about it – although it biases the sample, it is an almost universal bias, and thus permits comparison to the many many other, similarly-limited datasets (see Lees 2021 Neth J Sea Res, and Powell &Mann 2016 Cont Shelf Res for discussions). I especially admire that they acquired these samples on SCUBA, allowing for careful ad known placement… and 3 replicates per site.

R/ Agree. We added the following sentence in the Sampling section: “A potential caveat with this sampling is that it may miss deep burrowing species inhabiting below 10 cm deep”.

- Description of field study areas is exemplary – these authors really describe the areas in useful ways, including extent of physical disturbance and, of course, likely human effects, drawing on earlier work (includes the admirable Table 1)… thus well-chose field areas. The 0.5mm sieve is much finer than is usual (1mm, and in some live-dead studies by geologists/paleoecologists, a 2mm or coarser sieve is used) – only a concern because this likely included juvenile individuals that can be hard to identify AND have unlikely persistence, both as living individuals (persistence to adulthood) and as dead shells (postmortem) – although I appreciate that tropical faunas can have many many ‘micromollusks’ that might be largely missed with a 1 mm mesh. Just something worth mentioning, as a guide to other researchers thinking of repeating this kind of analysis elsewhere.

R/ Full agreement, we chose 0.5-mm mesh size based on previous reports about the huge diversity of micro-mollusks in Cuban waters, as there are many new species described by Espinosa et al. from both studied gulfs. 

- Live data have already been reported, in Armenteros et al 2018 – those results should be provided in tabular form here because the sample sizes of this living assemblage is critical to evaluating live-dead agreement.

R/ Noted. We removed objective 2 from the paper, therefore the data from Armenteros et al. are not needed any more in the paper.

- Analysis – Richness: I really like the authors’ approach for taxonomic richness, strongly prefer accumulation curves rather than post-hoc rarefaction fittings, AND I like using effective numbers of species for richness – super. Phylo richness – not familiar, feels like a stretch to add this, but no objection. Functional richness – good use of Todd & Mikkelson/Bieler for coding taxa; although the N of categories is (slightly) smaller than the S of species, could still use the same metrics; but I like bringing in these FD metrics from biogeography/macroecology. Beta – many ways to do it, they should also consider (and in discussion compare their results to) the methods used in Tomasovych/Kidwell 2009-2011 (e.g., diversity transfer from alpha to beta, spatial gradient capture, richness/evenness at alpha) – a series of papers by those authors (summarized in Kidwell Tomasovych 2013) explore the effects of time-averaging on the suite of metrics used here, seems to be one part of the paleo/taph literature missed. 

R/ We agree that there are many ways to approach this sort of data. This is in part the reason for a paragraph of the introduction devoted to the most relevant (in our opinion) and modern techniques for analysis of diversity. We do not wish to enlarge much the manuscript with another set of metrics and prefer to keep it as it is. Indeed, reviewer # 2 is requesting to shorten the manuscript’s length.

- Evenness – would prefer to see a quantitative metric of evenness, such as the sample-size-independent Hurlbert’s PIE, rather than only a graphical depiction; N singletons and doubletons is a measure of the proportion of rare taxa, which is useful/important, but not the same as dominance/evenness that focus on the proportional abundances of the top few taxa and overall shape of the cumulative richness curve. PIE is now widely used in live-dead analysis.

R/ We disagree. We prefer to use graphical devices (dominance curves) instead of reducing all the information to a single index. 

- Ah—structure = simper analysis, not what I’d call structure, but rather inter-sample similarities, series of pairwise comparisons…

R/ Agree. We change the term “structure” by composition, which is clearer.

- Comparison of live-dead – using the ancient, very simple metrics of Kidwell & Bosence 1991. That’s OK as long as include a table with the raw N’s being compared (e.g., how similar is the N of living and dead individuals?), because those ancient authors found that the small-sample size of the living assemblages in most studies was the most important factor explaining (correlated with) the low apparent live-dead agreements (which the later modeling of Kidwell and others showed was true). But a full live-dead nowadays would explore live-dead using standard ecological measures such as (dis)similarity of paired live-dead species lists for a site/habitat/region, ditto in rank-abundance of taxa live&dead, nmds etc. [I do note a NMDS lines469; another use of the word ‘structure’, here meant to indicate *spatial variation in species composition of assemblages, whereas structure also used to indicate RADs… be careful, would be good to sharpen language. **I fear that the omission of such measures owes to very low abundances of living individuals, an especially large challenge in tropical settings in my experience… typically we have to process far more sediment for living than for dead to achieve statistical power (see Powell et al from the early 1980s for the first time this was recognized and acted upon; Gilad et al 2018 did the same thing, sieving many-x more volume of sediment for live than dead, in tropical shelf habitats for the same reason). If this is indeed the situation, the authors, again, do not need to be defensive but simply straightforward and ‘up-front’, that is explain from the start that there are limits to what they can do analytically given extremely disparate live-dead sample sizes (and probably extremely small Ns of living in absolute terms, e.g. anything <50 inds and especially <20 inds).

R/ We removed the comparison live-dead, which corresponds to objective 2, because the very low abundance of living individuals, for most of the sites, abundance was < 20 individuals. We changed the term “structure”.

- Results – first para on general features is where the authors should immediately start differentiating their living and death assemblages – e.g., of these 7113 inds, how many dead & how many alive, ditto the S of species orders etc. If all of these numbers and others in the Results section are going to refer exclusively to the death assemblages, then this also needs to be made explicitly clear here, e.g. “All results refer only to dead-shell assemblages unless otherwise noted (e.g., section xx on dead vs total comparison).” I find it hard to proceed without knowing this basic information.

R/ We agree and follow your advice, so in the first paragraph of the results we mention that we identified 7113 dead individuals. 

- Line 333 – ‘fluctuations’ typically would refer to variability in time, whereas here I think the authors mean variability among spatial replicates at a site etc collected during the same season… watch for this.

R/ Agree. We changed fluctuation × variability.

- Line 514 Table 2 – Finally! Info on N live and N dead… and indeed the disparity is super high, not shocking for a tropical shallow marine system but very troubling for live-dead comparison. My suspicion re why the authors chose to compare dead versus live+dead are confirmed. This is not a scientifically fair/appropriate test of the ability of the dead to serve as a surrogate of the usual, entirely-live-collected kind of data that biologists typically have – and moreover flips the usual disparity (usually the ‘whole’ is much larger/broader/more inclusive sampling of the target than is the subset being tests as a ‘surrogate’/proxy of i). The results are of course interesting, but mostly for enlightening readers about the challenges of live-dead analysis in such settings. I think the discussion and conclusions from this section will thus need to be very modest, very modulated. ** For the editor: The disparities are e.g. 1500 dead inds versus 5 living ones, etc, with only two sites yielding >20 individuals, with 20 being an absolute minimum N of individuals to do any kind of nmds etc.

R/ Agree. We acknowledge that the disparity between the two kind of assemblages is too large for a fair comparison, and most of the sites has < 20 alive individuals. Thus, we removed from the paper the objective 2 about comparison live-death, that necessarily implies to change the title. However, we moved the information about the abundance and observed richness of dead individuals to the table 1.

- In this table 2, I’m most intrigued by the (to me) large number of taxa found exclusively alive, i.e. “alive-only” – I would not expect this given the huge disparity in sample size, i.e. that any taxa would not be encountered among the dead. 2 explanations = (1) new, invasive taxa, populations still small, little opportunity for dead shells to accrue to measurable levels in the death assemblage [given the species, I doubt this, but authors should (in the discussion) indicate if these are all natives or otherwise commonly encountered already by others in Cuba; or (2) taphonomic bias, i.e. these are all taxa with truly exceptionally low preservation potential, e.g. from their especially small/thin-shelled bodies, and/or super-rarity, and/or high-organic aragonitic shell types, and/or life-habit (e.g. I find that commensals have very low preservation potential). From my experience, Solemya has a super-low pres pot (as do lyonsiids, pinnids, some pandoriids), and I commonly find that Bittium, Caecum, and Rissoids/small parasitic & minute herbivores are strongly underrepresented in the dead (but not actually absent) Limaria and Americardia & Codakia should definitely not fall into this category… anyway, a surprising and super-interesting result

R/ We removed this part of the paper, although we added the abundance and number of observed species of dead mollusks in the table 1, as basic numbers for diversity assessment and sample size information.

- 523 contribution of shelf species to deep-sea death assemblages – wonderful, super useful observations. It would be super-helpful to provide the typical or maximum known body sizes of these taxa found off shelf, and/or, even better, the typical or max body sizes observed among thte individuals actually collected. Would make a more powerful discussion/conclusions

R/ These observations are valuable, but as indicated by reviewer # 2, the occurrence of a single deep-sea site (with only data of presence/absence) is a main weakness of this part. We preferred to remove objective 3 from the paper.

- Table 3 this section – Please provide info at the bottom of each column on the total N of the death assemblage so that readers can get full value from these prop abundance values. E.g. does 100% (“1”) mean 10 out of 10, 100 out of 100 inds, or 1 out of 1? The data on raw N of dead shells from a tropical slope would be valuable in itself.

=Interesting from a taphonomic perspective that all are gastropod species w one exception… because epifaunal, which is long thought to increase the potential for postmortem transport (because they die above the sediment-water interface—idea probably in KidwellBosence)? Some info on life habit, not just gastro-bivalve class membership, would be nice to add to this table, or at least to bring into discussion.

R/ We removed table 3 from the paper.

Discussion 

---

## [Decision Letter · Decision Letter 1]

15 Mar 2024

PONE-D-23-26159R1Taxonomic, phylogenetic, and functional diversity of mollusk death assemblages in coral reef and seagrass sediments from two shallow gulfs in western Cuban archipelagoPLOS ONE

Dear Dr. Peraza-Escarrá,

Thank you for submitting your manuscript to PLOS ONE. After careful consideration, we feel that it has merit but does not fully meet PLOS ONE’s publication criteria as it currently stands. Therefore, we invite you to submit a revised version of the manuscript that addresses the points raised during the review process.

We look forward to receiving your revised manuscript.

Kind regards,

José A. Fernández Robledo, Ph.D.

Academic Editor

PLOS ONE

Journal Requirements:

Additional Editor Comments:

Please address the comments of the second reviewer.

Best,

-j

Reviewers' comments:

Reviewer's Responses to Questions

**Comments to the Author**

1. If the authors have adequately addressed your comments raised in a previous round of review and you feel that this manuscript is now acceptable for publication, you may indicate that here to bypass the “Comments to the Author” section, enter your conflict of interest statement in the “Confidential to Editor” section, and submit your "Accept" recommendation.

Reviewer #2: All comments have been addressed

Reviewer #3: (No Response)

2. Is the manuscript technically sound, and do the data support the conclusions?

Reviewer #2: Yes

Reviewer #3: Yes

3. Has the statistical analysis been performed appropriately and rigorously? 

Reviewer #2: Yes

Reviewer #3: Yes

4. Have the authors made all data underlying the findings in their manuscript fully available?

Reviewer #2: Yes

Reviewer #3: Yes

5. Is the manuscript presented in an intelligible fashion and written in standard English?

Reviewer #2: Yes

Reviewer #3: No

6. Review Comments to the Author

Reviewer #2: The manuscript titled "Taxonomic, phylogenetic, and functional diversity of mollusk death assemblages in coral reef and seagrass sediments from two shallow gulfs in the western Cuban archipelago," authored by Peraza-Escarrá and collaborators, evaluates diversity patterns of mollusk assemblages in two gulfs and across two different habitats. This topic is of significant interest to the scientific community. Following the authors' thorough addressing of all raised issues from the previous review, as well as the resolution of my concerns, the manuscript has undergone substantial improvement. In my assessment, it has now reached the necessary standard for acceptance for publication

Reviewer #3: This manuscript addresses the diversity (taxonomic, phylogenetic and functional) of mollusc death assemblages in two different habitats in the Western Cuban Archipelago.

The topic is interesting, worth of being published and the manuscript seems to have been well improved after the first round of review. The manuscript is well organized and the analyses are appropriate for the goals of this work. I propose some suggestions aimed at improving the clarity and importance of the work, most of my comments are in the annotated text. Some additional points are below.

1. It would be helpful to better explain why phylogenetic diversity is important - for Cuba & the studied environments - both in the introduction and the discussion.

2. Introduction: it needs to focus more on the problems that this paper is addressing, the Cuban archipelago and the gap in knowledge that this work fills.

3. Figure 9: please change the position of the text so that the letters don't overlap

4. Figures 6, S1, S2 and S4: "sp." should not be in italics

5. Paragraph starting at line 594: what is the relation between the information from the literature and the present work?

6. Lines 638-646: please rephrase for clarity

7. Time-averaging (lines 649-666): this part can be reduce, the information on the subject can be provided in the references used; it would be better to focus on the time-averaging of the studied material.

8. Discussion: refer to the results and the figures that are the main focus of the paper

9. Conclusions: the section can start with a sentence on the present work, so as to present a global view. The section should highlight the importance of the work, what new it brings rather than summarizing the main points of the results and discussion.

Lastly, the English language should be revised throughout the document.

7. PLOS authors have the option to publish the peer review history of their article (what does this mean?). If published, this will include your full peer review and any attached files.

Reviewer #2: No

Reviewer #3: **Yes: **Danae Thivaiou

---

## [Author Response · Author response to Decision Letter 1]

10 Apr 2024

Response to Editor:

Dear editor, thank you very much for guiding us during the revision process of our manuscript. As requested, we have addressed all the comments made by the Reviewer #3. We also revised the English usage and improved the readiness of some parts of the manuscript.

Response to reviewers:

Dear reviewers, thank you very much for taking the time to revise our manuscript, we appreciate your comments and criticisms very much. We have addressed all your comments. Please, see our responses below.

Reviewer #2:

The manuscript titled "Taxonomic, phylogenetic, and functional diversity of mollusk death assemblages in coral reef and seagrass sediments from two shallow gulfs in the western Cuban archipelago," authored by Peraza-Escarrá and collaborators, evaluates diversity patterns of mollusk assemblages in two gulfs and across two different habitats. This topic is of significant interest to the scientific community. Following the authors' thorough addressing of all raised issues from the previous review, as well as the resolution of my concerns, the manuscript has undergone substantial improvement. In my assessment, it has now reached the necessary standard for acceptance for publication.

R/ Thank you very much for your valuable comments that significantly improved our paper. 

Reviewer #3:

This manuscript addresses the diversity (taxonomic, phylogenetic and functional) of mollusc death assemblages in two different habitats in the Western Cuban Archipelago.

The topic is interesting, worth of being published and the manuscript seems to have been well improved after the first round of review. The manuscript is well organized and the analyses are appropriate for the goals of this work. I propose some suggestions aimed at improving the clarity and importance of the work, most of my comments are in the annotated text. Some additional points are below.

R/ Thank you very much for your valuable comments that significantly improved our paper. For the comments below, we provide a response point by point. We corrected most of the comments annotated in the PDF; however, we disagree in a few comments and also explained here our points of view.

1. It would be helpful to better explain why phylogenetic diversity is important - for Cuba & the studied environments - both in the introduction and the discussion.

R/ Agree. We added the following sentence in the introduction: “Phylogenetic diversity adds an evolutionary perspective to the biodiversity patterns …”. In the discussion and conclusions, we reinforced the component of phylogenetic diversity. 

2. Introduction: it needs to focus more on the problems that this paper is addressing, the Cuban archipelago and the gap in knowledge that this work fills.

R/ Agree. We rephrased the paragraph devoted to the studies done in Cuba and highlighted the knowledge gaps.

3. Figure 9: please change the position of the text so that the letters don't overlap.

R/ Agree. We already moved the labels, but we wrongly uploaded an old version of this figure. We now upload the right version.

4. Figures 6, S1, S2 and S4: "sp." should not be in italics

R/ Agree. We fixed the issues.

5. Paragraph starting at line 594: what is the relation between the information from the literature and the present work?

R/ Agree. We modified the paragraph.

6. Lines 638-646: please rephrase for clarity

R/ Agree. We improved the wording.

7. Time-averaging (lines 649-666): this part can be reduce, the information on the subject can be provided in the references used; it would be better to focus on the time-averaging of the studied material.

R/ We agree. The two paragraphs were modified and fused.

8. Discussion: refer to the results and the figures that are the main focus of the paper.

R/ Agree. We added references to tables and figures where necessary. 

9. Conclusions: the section can start with a sentence on the present work, so as to present a global view. The section should highlight the importance of the work, what new it brings rather than summarizing the main points of the results and discussion.

R/ Agree. We modified the conclusions, removing the repetition of results and highlighting the importance of our work.

10. Lastly, the English language should be revised throughout the document.

R/ Agree. We extensively revised the English usage.

Two comments in the annotated PDF that we disagreed with and further explain here our point of view.

- Please explain why you did this. Usually, for all bivalved organisms (bivalves, ostracods, brachiopods etc), we divide the total number by two, in order to have a better estimate of the real number of specimens. See Kowalewski et al. 2002 (https://doi.org/10.1130/0016606(2002)114<0239:MHAOMM>2.0.CO;2) "However, because an individual bivalve yields two skeletal elements (valves) and, thus, is twice as likely to be sampled as an individual gastropod, which produces only one shell, all analyses that compare abundance of bivalves and gastropods require a correction: The number of bivalve valves has to be divided by 2".

R/ The suggested approach is better used when the objective is an accurate assessment of abundance, usually to compare bivalves versus gastropods (as Kowalewski et al. 2002 stated). Our approach is also valid because the goal is to assess diversity and in much lesser extension to estimate the absolute abundance. This approach is also known as “Maximum number of individuals approach” (Gilinsky and Bennington 1994). Additionally, when dividing the number of counted valves by two, we would artificially inflate those diversity metrics that depend on the abundance, for instance the rarefaction and the evenness. We consider that our approach is correct, and no change has been made.

Gilinsky, N. L., & Bennington, J. B. (1994). Estimating numbers of whole individuals from collections of body parts: a taphonomic limitation of the paleontological record. Paleobiology, 20(2), 245-258.

- better use the term sessile. Can you also please add the name(s) one or two species here? This can be done also for the other functional groups (chemosynthetic deposit feeders, herbivores etc.).

R/ We think that sedentary is more appropriate than sessile as they are not the same. We keep the category sedentary. However, we changed the term immobile × sessile, as indicated in another comment.

We also think that is not necessary to mention the names of the species. These names can be found in the S1 table. But more important when using the functional approach, we do not want to go back to the species identity, instead the objective is to find patterns based solely on the trait identity.

---

## [Decision Letter · Decision Letter 2]

26 Apr 2024

Taxonomic, phylogenetic, and functional diversity of mollusk death assemblages in coral reef and seagrass sediments from two shallow gulfs in western Cuban archipelago

PONE-D-23-26159R2

Dear Dr. Peraza-Escarrá,

We’re pleased to inform you that your manuscript has been judged scientifically suitable for publication and will be formally accepted for publication once it meets all outstanding technical requirements.

Kind regards,

José A. Fernández Robledo, Ph.D.

Academic Editor

PLOS ONE

Additional Editor Comments (optional):

Reviewers' comments:

Reviewer's Responses to Questions

**Comments to the Author**

1. If the authors have adequately addressed your comments raised in a previous round of review and you feel that this manuscript is now acceptable for publication, you may indicate that here to bypass the “Comments to the Author” section, enter your conflict of interest statement in the “Confidential to Editor” section, and submit your "Accept" recommendation.

Reviewer #3: All comments have been addressed

2. Is the manuscript technically sound, and do the data support the conclusions?

Reviewer #3: Yes

3. Has the statistical analysis been performed appropriately and rigorously? 

Reviewer #3: Yes

4. Have the authors made all data underlying the findings in their manuscript fully available?

Reviewer #3: Yes

5. Is the manuscript presented in an intelligible fashion and written in standard English?

Reviewer #3: Yes

6. Review Comments to the Author

Reviewer #3: Thank you for taking into account the previous comments, the manuscript has been greatly improved and is now ready to be accepted. Few very minor comments are in the text.

7. PLOS authors have the option to publish the peer review history of their article (what does this mean?). If published, this will include your full peer review and any attached files.

Reviewer #3: **Yes: **Danae Thivaiou (Naturhistorisches Museum Basel)

---

## [Editor Report · Acceptance letter]

1 May 2024

PONE-D-23-26159R2 

PLOS ONE

Dear Dr. Peraza-Escarrá, 

I'm pleased to inform you that your manuscript has been deemed suitable for publication in PLOS ONE. Congratulations! Your manuscript is now being handed over to our production team.

Kind regards, 

on behalf of

Dr. José A. Fernández Robledo 

Academic Editor

PLOS ONE